# Protecting Traditional Knowledge through Biocultural Community Protocols in Madagascar: Do Not Forget the "B" in BCP

**Manohisoa Rakotondrabe * and Fabien Girard** 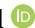

CRJ, Faculty of Law, University of Grenoble Alpes, 38400 Grenoble, France; fabien.girard@univ-grenoble-alpes.fr
* Correspondence: manohisoa.rakotondrabe@univ-grenoble-alpes.fr

**Abstract:** As in many other countries in the south, the traditional knowledge (TK) of local communities in Madagascar is facing extinction. Biocultural community protocols (BCP), introduced in Madagascar following the implementation of the Nagoya Protocol (2010) and defined by the Mo'otz Kuxtal Voluntary Guidelines as "a wide range of expressions, articulations, rules and practices produced by communities to indicate how they wish to engage in negotiations with stakeholders", holds out hopes for TK protection. By analysing two pilot BCPs in Madagascar, one established around the *Motrobe* (*Cinnamosma fragrans*) with a view to strengthening the existing value chain (BCP in Mariarano and Betsako) and the second initially established around plant genetic resources for food and agriculture (BCP of the farmers in Analavory), this study aims to assess the place and value ascribed to TK in the overall BCP development process and to analyse whether or not the process has helped to strengthen and revitalise TK at the community level. The ethnographic studies show commonalities in both BCP, in particular their main focus on access and benefit-sharing mechanisms, this against the backdrop of an economic model which stresses the importance of financial and institutional incentives; and conversely, a relative disregard for what relates to the biocultural dimension of TK. Local taboos (*fady*) as well as traditional *dina* (social conventions), which have long allowed for the regulation of access to common resources/TK, are scarcely mentioned. Based on these findings, we conclude that in order to revitalise TK, the process of developing BCPs should recognise and give special importance to TK, considering it as a biocultural whole, bound together with the territory, local customs, and biological resources; or else, TK is likely to remain a commodity to be valued economically, or a component like any other.

**Keywords:** Madagascar; traditional knowledge; biocultural community protocol; development process; genetic resources; phytogenetic resources for food and agriculture; local community right

## 1. Introduction

"Traditional knowledge" (TK) or "traditional ecological knowledge" (TEK) is what Berkes defines as "a cumulative body of knowledge, practice and belief, evolving by adaptive processes and handed down through generations by cultural transmission" [1]. Moreover, "as a knowledge–practice–belief complex, indigenous knowledge includes an intimacy with local land, animals, and plants. It also includes institutions (rules and norms) about interacting with the environment, and it includes worldview, as the worldview shapes the way people make observations, make sense of their observations and learn. These levels of ecological knowledge may overlap, change over time, and interact with one another" [2]. TK is therefore rooted in and supported by worldviews that connect the relationships between humans and nonhumans in a specific way, and constantly evolve according to reconfigurations in these relationships.

In Madagascar, although TK was considered to be against God's will from the time of colonisation and evangelisation, and then as an impediment to development when the country gained its independence, it continued to be used and transmitted from generation

to generation by local communities in order to manage social conflicts, to maintain social security, and to regulate the use of complex socio-ecological systems. This is clearly visible through the persistence of *dina*, or traditional social conventions, which are endogenous to communities. It is also visible through the taboos or *fady*, which continue to govern life in many places. A *fady* generally refers to a constraint on a particular activity in a specific location [3]. It may be limited to a particular family or clan, but does not necessarily affect the entire community [4]. Still strictly respected by the *fokonolona* (the entire community, defined as a "set of people, households living in the same space, which can be linked to it by their subsistence activities; a lineage; a set of people linked to each other by their way of life depending on natural resources" [5]) *dina*, *fady* and *fihavanana* (a set of rules and norms that define a code of good conduct in Malagasy society [6,7]) form part of the pillars of the self-regulatory social system, which governs the community's use of natural resources.

The Grande Île is a biodiversity hotspot, home to a treasure trove of endemic riches that have too often been plundered, to the great contempt of local communities holding the TK, which is intrinsically linked to these riches. The case of the Madagascar periwinkle (*Catharanthus roseus*) is, without a doubt, the most emblematic example that has served to illustrate discussions around biopiracy, defined as "the unauthorised commercial use of biological resources and/or associated traditional knowledge, or the patenting of spurious inventions based on such knowledge, without compensation" [8], it and continues to mark Malagasy politics in terms of biodiversity [9–11]. Madagascar also stands out for the singular place it occupies due to its natural and intangible resources, its colonial history, its state of economic development, and in terms of its experimentation with new global environmental policies and new instruments to protect biodiversity, the latest incarnation of which are biocultural community protocols (BCPs).

BCPs are part of a rising idea of biocultural diversity and "stewardship", and recognise "the holistic interconnectedness of humanity with ecosystems and obligations and responsibilities of indigenous and local communities, to preserve and maintain their traditional role as traditional guardians and custodians of these ecosystems through the maintenance of their cultures, spiritual beliefs and customary practices" [12].

They made their way into Madagascar law (as in the case with many other countries in the global south) through the Nagoya Protocol, which was ratified by the country in 2014 and principally aims to implement the third goal of the Convention of Biological Diversity (CBD) of fair and equitable sharing of benefits deriving from the use of genetic resources. The Nagoya Protocol strengthened the mechanism to combat the illegitimate appropriation of IPLCs' genetic resources and TK [13]. In particular, it sets out criteria to promote and protect TK and the "prior informed consent or approval and involvement of indigenous and local communities" (Nagoya Protocol, Art. 6 para. 3(f) and Art. 7) [ . . . ] The "mutually agreed terms" (MAT) must then ensure that benefits "are shared in a fair and equitable way" (Art. 5, para. 2, Art. 6, para. 3, (g), and Art. 7) [14] taking into consideration "indigenous and local communities' customary laws, community protocols and procedures, as applicable, with respect to traditional knowledge associated with genetic resources" (Art. 12, para. 1). The Nagoya Protocol also introduced biocultural community protocols (BCPs), referred to as "community protocols", into the arsenal of instruments to preserve biodiversity and protect IPLCs (Nagoya Protocol, Art. 12, para. 1, para. 3(a) and Art. 21(i)). According to the accompanying guidelines, community protocol is defined as "a broad array of expressions, articulations, rules and practices generated by communities to set out how they expect other stakeholders to engage with them [15]." Protocols may also "( . . . ) reference customary as well as national or international laws to affirm their rights to be approached according to a certain set of standards". They may also give communities an opportunity to "( . . . ) focus on their development aspirations visa-vis their rights and to articulate for themselves and for users their understanding of their bio-cultural heritage ( . . . )" [15].

BCPs can be described as charters that set out or codify the rules and procedures by which a community normally manages its resources and associated TK, and regulates access

to them. According to their advocates, BCPs are depositories of customary traditions and customary rules for managing the tangible and intangible heritage of local communities, in addition to being political tools to advocate for better recognition of their rights over their land and culture.

BCPs have given rise to a great deal of hope among defenders of IPLCs, who have seen them as an effective means of supporting and protecting their TK. As Brendan Tobin noted, "community protocols have been seen as 'one of the most effective tools for securing effective [TK] protection' as they can bridge customary, national and international law, leading to community protocols becoming a useful aid in the regulation and protection of traditional knowledge" [16].

This article focuses on the study of the application of these BCPs in the Malagasy context, asking the principal question of whether BCPs really have enabled the creation of a framework conducive to the development and protection of the TK of local communities.

Just one year after the ratification of the Nagoya Protocol in 2015, Madagascar had already established three BCPs, two of which are at the heart of our study. The first is the BCP of the seven communities of Mariarano and Betsako, established under the Environmental Management and Support Programme (referred to by its French acronym, PAGE, and under the auspices of the GIZ) and focussed on promoting a forestry value chain (*Motrobe* or *Cinnamosma fragrans*) in the Boeny Region (see Scheme 1). The second is the Analavory farmers' BCP, established as part of a project funded by the Darwin Initiative, entitled "Mutually supportive implementation of the Nagoya Protocol and Plant Treaty" (see Scheme 2).

Madagascar is thus one of the first countries to adopt biocultural protocols and to recognise them in its regulatory framework. It is also one of the few countries to have developed protocols under two different projects, the first of which accompanied the deployment of the new access and benefit sharing framework in the country.

We can thus see the potential interest of comparing these two BCPs, one of which relies upon the second international text which makes up the international regime on access to genetic resources and benefit sharing (ABS) [17]: the International Treaty on Plant Genetic Resources for Food and Agriculture (ITPGRFA), also ratified by Madagascar in 2006. In contrast to the CBD/Nagoya Protocol regime, the ITPGRFA only covers a subcategory of genetic resources, known as "plant genetic resources for food and agriculture" (PGRFA). PGRFAs are subject to specific regime of "facilitated access". Where the CBD and the Nagoya Protocol still turned to bilateral negotiations between the provider and the beneficiary of a resource—necessarily resulting in an access and benefit sharing (ABS) agreement—the ITPGRFA established a multilateral system (MLS) which is a sort of virtual basket covering 64 cultivated species with regard to which the parties agree to grant facilitated access via a standard material transfer agreement (SMTA). Article 9 of the ITPGRFA recognises the contributions made by local and indigenous communities, as well as farmers, to the conservation and sustainable use of PGRFAs.

While the Mariarano and Betsako BCPs fall relatively traditionally within the regime complex made up of the CBD/Nagoya Protocol, the Analavory farmers' BCP is a little different as it covers both genetic resources and PGRFAs and raises the issue of the connection between the Nagoya Protocol and the ITPGRFA. These two Malagasy examples also reveal another notable characteristic: the very low integration of biocultural rights and biocultural heritage, and therefore the near complete absence of references to customary rights, land rights, and local normative authority. Importantly, BCPs are inextricably linked to "biocultural rights", which are characterized as a "bundle" or "basket" of rights consisting of: (i) the right to land, territory and natural resources; (ii) the right to self-determination, principally understood here in its "internal" dimension, i.e., the right of communities to autonomy and to self-administration; (iii) cultural rights. In an original way, the bundle includes (iv) a duty of "stewardship", which arises from the "ethic of stewardship" associated with the practices, values and lifestyles of local populations [18,19].

Their content reveals a sort of tropism which encourages the economic development of genetic resources and TK, which, while consolidating it, implies a notable reductionism: TK and local resources are transformed into "raw inputs" [20], in other words, a commodity, thus severing the link between land, culture, spirituality, and traditional regulation [21]. It is certainly true that BCPs may ensure, as their advocates promise, an "incomplete commodification", inspired as they are in this regard by Margaret Radin's proposals [22]. The idea is that, while introducing IPLCs' TK and traditional resources into international trade markets, the market inalienability of certain elements of their tangible and intangible heritage—elements (such as language, land, and culture) upon which the wellbeing and identity of the community depend—could be maintained [18,21,22].

However, in the detail, this implies comprehensive work in developing BCPs, particularly involving reflection upon the way in which the alienable part (that which is to enter the market) is connected to the rest of the biocultural heritage, which must remain *extra commercium*.

How did this work out in the context of the two Malagasy BCPs? In concrete terms, how were TK taken into account in the development process of the Analavory and Mariarano and Betsako BCPs?

Our study reveals that if a holistic approach that maintains the position of TK within the biocultural heritage (local customs, land, and identity) is lacking, there is a risk that this knowledge will be treated as an "array of 'raw' inputs for life science corporations" [20], or as goods [20], with a certain number of possible consequences on community dynamics and social reproduction. We also suggest a few avenues for better integrating the "biocultural" dimension into the development of future BCPs, in Madagascar and elsewhere.

After a description of the research methodology and general context of the two BCPs, our analysis will turn to the development process of the two BCPs, with a view to examining the way in which they take into account the biocultural heritage and protection of TK. The discussion part will first look at the factors, which may have hindered the BCP development process and may have had an impact on the consideration of biocultural heritage. We will then look at the possible avenues that could be explored to ensure a revitalisation of TK through BCPs by repositioning them within their biocultural context. The main ideas and inputs of the study are outlined in the conclusion, with insights into the relevance of the debate on BCPs for biocultural-based initiatives in the "Global North".

## 2. Materials and Methods

The main resources used in the study are the two BCPs as written sources jointly produced by the communities involved and the NGOs, which supported their development.

The first of these is the BCP produced by the seven local communities managing the forestry resources in the communes of Mariarano and Betsako (the Boeny Region) (see Scheme 1). These two communes are around 90 km from Majunga and belong to the District of Majunga II in the northwest of the island. They are inaccessible by car during the rainy season (especially from October to April). Mangroves, raffia, and successions of shrubby savannahs and dry deciduous forests with a high rate of endemism (such as a substantial part of the Didiereaceae and Fabaceae families, the Andasonia, which includes six species of baobab indigenous to the region) [23] contribute to the region's biodiversity. The Sakalava people, a local ethnic group, have long had strong relationships with their environment, believing that the forest has a sacred dimension due to its divine origin. It is "the spirits' favorite home and a haven for many creatures eager to punish anyone who enters without first declaring himself" [24]. There are also *fady* days, i.e., days when no one is allowed to work in the field or enter the forest (e.g., Thursday, described as the day of the *tsiny* (See below [25] (p. 17)), at the risk of attracting the anger of the ancestors. Certain animals (such as the crocodile) and trees (such as baobabs and tamarind trees) are sacred and where rituals are performed [26]. Tamarind trees can also serve as the tree under which important village meetings are conducted [27] (Figure 1). However, due to illicit extraction of timber and wood energy and slash-and-burn activities (*hatsaky*), natural

resources in these areas are deteriorating at an alarming rate. A few forest islands have remained unaltered; they are remains of woodlands where the Sakalava people constructed *doany* [28] or family tombs, which are revered as sacred sites and are strictly protected by local customs (*fady*).

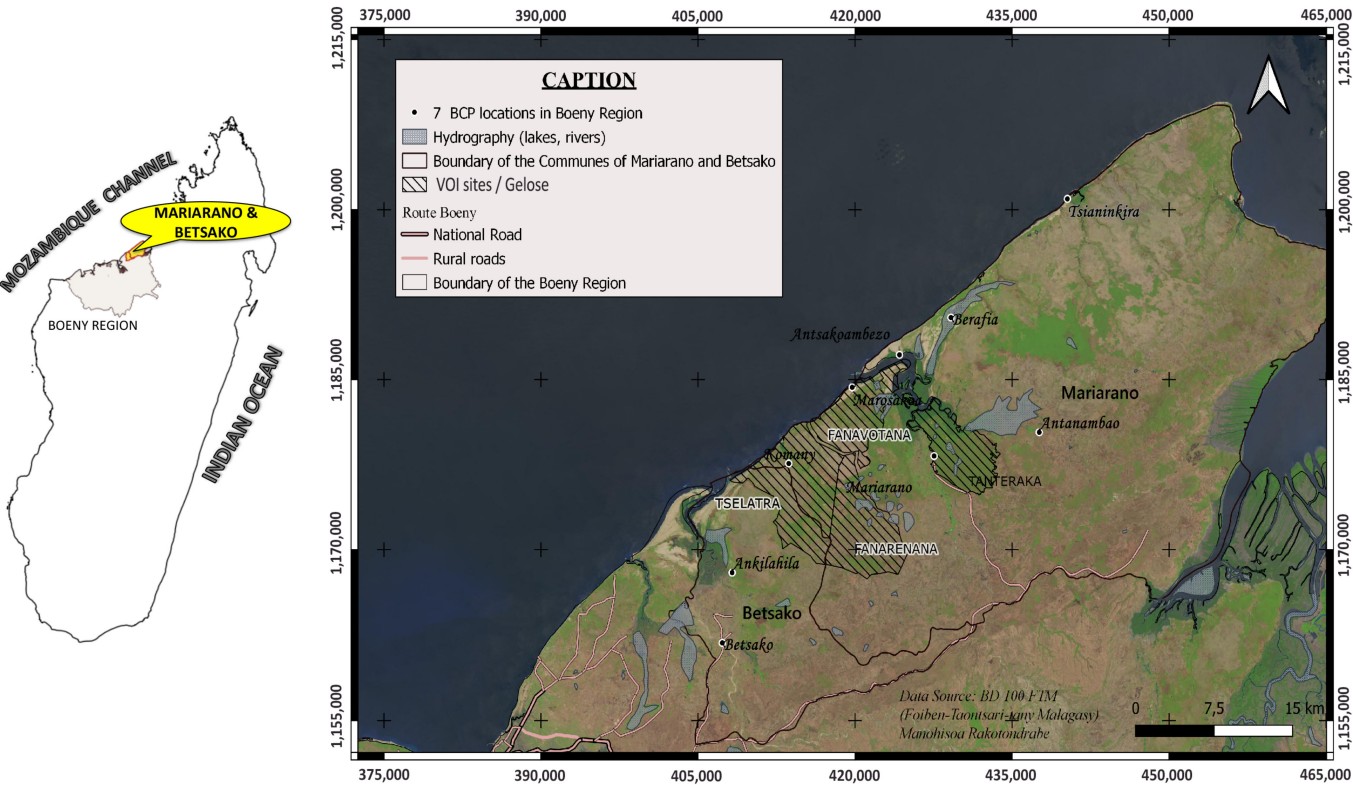

**Scheme 1.** Location of Mariarano and Betsako BCP (© Manohisoa Rakotondrabe).

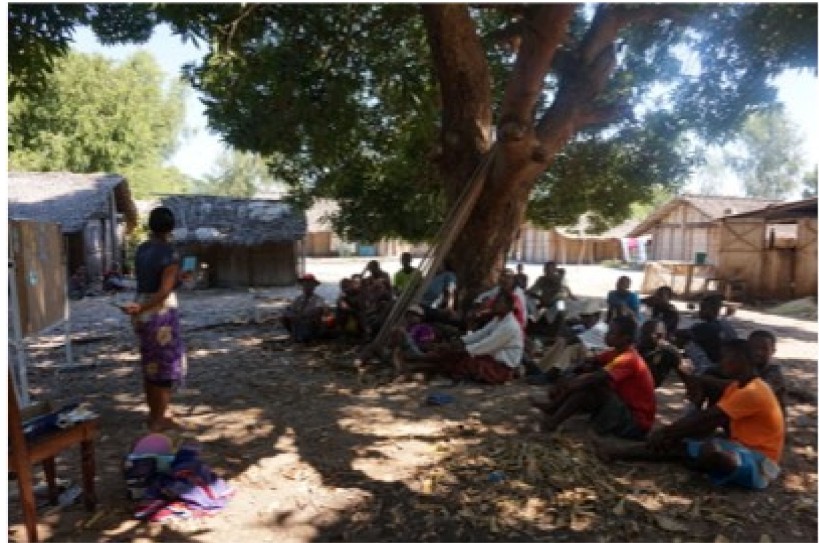

**Figure 1.** Village meeting on BCP organised by GIZ and Natural Justice in Ankilahila (Commune of Mariarano) (© Jazzy Rasoloarijaona).

The BCP is entitled "Biocultural Community Protocol (BCP) of the local communities who are the custodians of biodiversity and the holders of traditional knowledge in Mariarano, Antanandava, Komamy, Ankilahila, Marosakoa, Tsakoambezo and Tsiankira"

(hereinafter referred to as the "Mariarano and Betsako BCP"). Established as part of the GIZ Environmental Management Support Programme, it consists of a 26-page document drafted in Malagasy by members of the local community (then translated into French), with the support of Natural Justice, a South African NGO working for defending the rights of indigenous populations and local communities in countries in the south, particularly in Africa. This NGO was founded in South Africa by Kabir Bavikatte and Harry Jonas, two international lawyers who played a decisive role in the emergence of biocultural rights and international recognition of BCPs by communicating about the BCPs around the world [29]. Natural Justice has a specific agenda in terms of ABS and the defense of indigenous and local community rights, and it was also involved in a Darwin Initiative-funded project that resulted in the construction of two more BCPs in Madagascar (in Analavory and Ampangalantsary) [30].

The process of negotiating and drafting the document began in August 2015 and lasted over two years, culminating in the signature of a first operational version of the BCP in November 2017 (see Figure A1). The process involved both preliminary studies to identify the potential value chains for establishing ABS mechanisms, but also to identify relevant local communities interested in the project. Community workshops and regional multi-actor workshops involving representatives of the seven communities, the *Sojabe*—village elders and traditional chiefs who [31] are the custodians of customs and memories [32], endowed with significant moral and religious powers and authority [33], the regional managers of the forestry administration and private operators working in the area, and particularly those involved in the *Motrobe* value chain. This plant, also known as *Mandravasarotra*, is endemic to Madagascar, and has long been used for medical purposes by local communities. The leaves are harvested and processed into essential oils in the Boeny Region before being exported.

The second BCP is that of the farmers of Analavory (Itasy Region; see Scheme 2), the title of which is the "Biocultural Community Protocol of the farmers of Analavory on access and sharing of benefit arising from the use of genetic resources and associated traditional knowledge" (hereinafter referred to as the "Analavory BCP"). The BCP in question was named after the rural commune of Analavory, which is located in the district of Miarinarivo, almost 100 km from Antananarivo, the capital of Madagascar. Analavory had forests of *zamborizano* (*Eugenia sakalavum* H.), an endemic species that was still abundant there until the 1970s. This plant was coveted for its ability to produce indigenous rum, especially used during social events (*famadihana*, circumcision, or funeral rites). The elders said: "*Mitovy amin'ny tenantsika ihany ny fitsinjovantsika ny tontolo iainana*"—"we maintain the forest as we preserve our bodies". However, the arrival of many migrants in the 1990s, as well as the fact that young people and newcomers have little respect for tradition, swiftly changed the surviving woodland remnants into sites for cultivation and homes.

The commune is home to around 68,000 inhabitants and has always been an area of high immigration, due in part to its geographical location and agricultural potential. The area is located at the crossroads of the RN1, a major, well-maintained road leading to the capital, and the RN23, which provides access to the tourist area of Ampefy and the irrigated areas suitable for rice cultivation in the district of Soavinandriana. Analavory is a productive farming area supplying Antananarivo with rice and market garden produce.

Established within the context of the Darwin Project, it was also facilitated by Natural Justice, which was mandated by Bioversity International in 2016 to support the process of developing BCPs in Madagascar and Benin. The Analavory BCP is a 27-page document written in Malagasy, drafted by certain members of the community. A French translation was also produced to meet the technical requirements of the project. For this BCP, the negotiation and drafting process lasted two years (2016–2017). It involved around 10 information workshops and village meetings before a first official version was signed on 27 December 2017 by the mayor of the commune and two regional representatives of the ministries directly concerned: the Ministry for Agriculture, Farming and Fisheries (MAEP) and the Ministry for the Environment and Sustainable Development (MEDD).

Natural Justice, the local community, and the project steering committee (which included the ITPGRFA and Nagoya Protocol National Focal Points in Madagascar and Benin, the ABS Initiative, the ITPGRFA and Nagoya Protocol Secretariat, and Biodiversity Network International, which served as the coordinator) agreed that the signed version would not be the final version and would, therefore, not be published, but that it would nevertheless allow the communities, their genetic resources, and TK to be protected in the event that negotiations were initiated with bioprospectors. Against the advice of Natural Justice, Bioversity International published this interim version online, which illustrates the conflictual relationship between the project leader and the subcontractor, an issue to which we will return later.

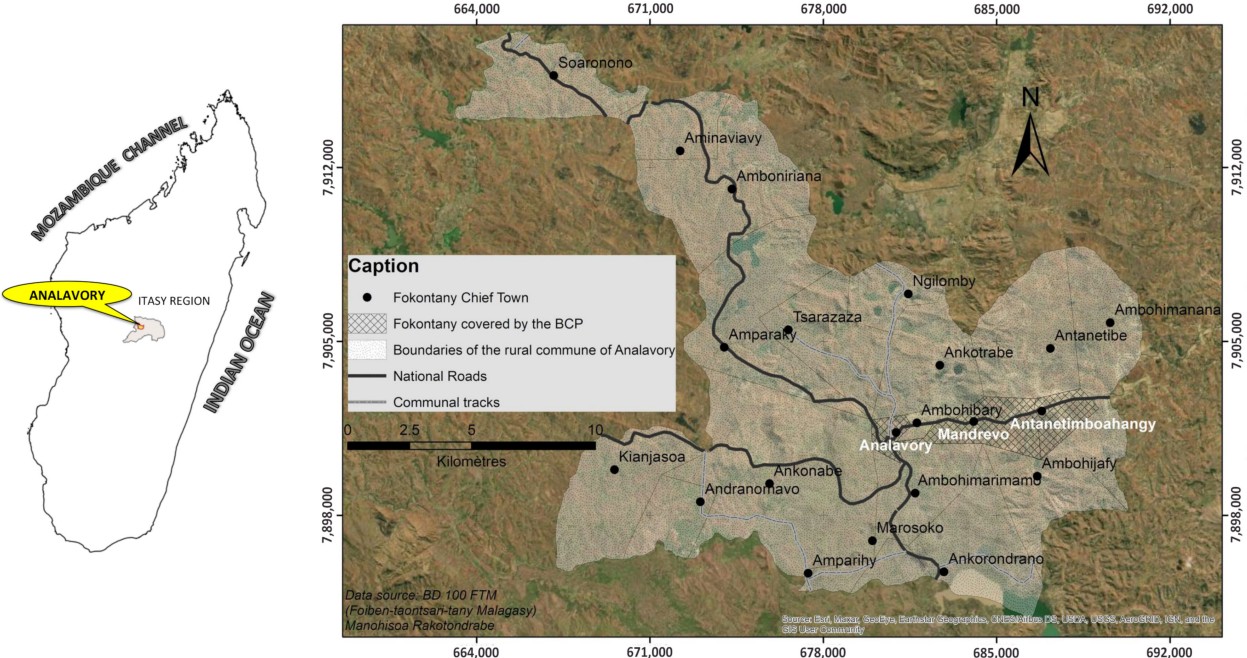

**Scheme 2.** Location of the Analavory BCP (© Manohisoa Rakotondrabe).

In addition to the Analavory BCP, the project supported the establishment of a seed bank and a community seed register. The community seed bank is a building that stores different varieties of seeds produced in the local area (see Figure 2a). The register, meanwhile, records the names and characteristics of each variety that is locally cultivated. Both are managed by a seed production cooperative, known as the FaMA Cooperative (*Famokarana Masomboly eto Analavory*), established during the project and which includes members of the Seed Producer Group (SPG)—i.e., groups of farmers who specialise in the production of seeds for marketing—as well as individual members of the community interested in the Darwin Project (see Figure 2b) [34]. During the process of setting up the BCPs, three *fokantany* were significantly involved: the *fokantany* of Mandrevo, Analavory, and Antanetimboahangy (See Scheme 2). The seed bank is located in the Mandrevo *fokantany*.

In terms of method, it should first be noted that the study of the two malagasy protocols is part of a larger comparative project (the "Bioculturalis" research project [35]) that involves the systematic study of the protocols included in the following table, with complementary ethnographic work in Madagascar and Panama (see Table 1).

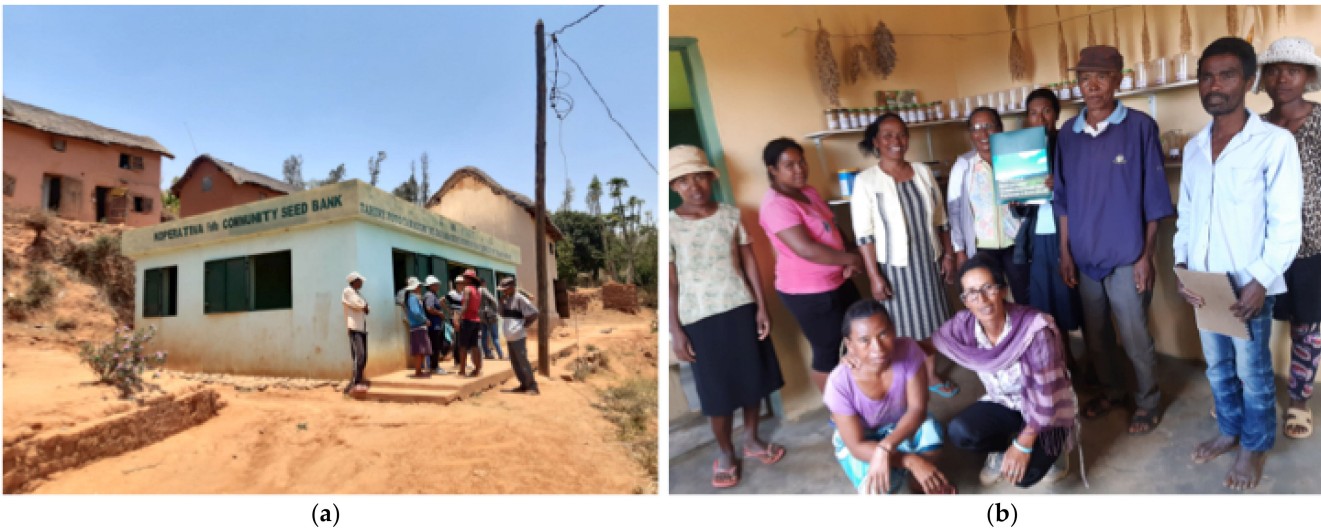

(**a**)                                              (**b**)

**Figure 2.** (**a**) Community seed bank in Mandrevo fokontany (© Manohisoa Rakotondrabe); and (**b**) Members of the FaMA cooperative and the seed collection samples kept in the seed bank (© Manohisoa Rakotondrabe).

**Table 1.** BCPs covered by the Bioculturalis research project (© Fabien Girard).

| Country | Biocultural Community Protocol | Source |
|---------|-------------------------------|--------|
| **Benin** | BPC of the Municipality of Tori-Bossito | [36] |
| | BCP of the community of Degbe Aguinninnou | [37] |
| **Kenya** | Ogiek Bio-Cultural Community Protocol | [38] |
| | The Lamu County Biocultural Community Protocol | [39] |
| **Madagascar** | BPC of Mariarano and Betsako | Not officially published |
| | BPC of the farming communities of Analavory | Not officially published |
| **Mexico** | BCP of Capulálpam de Méndez, Oaxaca | [40] |
| | Ek Balam BCP | [41] |
| **Panama** | Biocultural Protocol «Protection of the indigenous knowledge associated to genetic resources», El Piro Community, Ngäbe—Bugle Region, Panama | [42] |

The choice of Madagascar and of these two protocols in particular, one of which relates to the ITPGRFA and the other more classically to the ABS framework of the CBD/Nagoya Protocol, was approved at the outset of the project due to the recent legal recognition of BCPs in Madagascar and the existence of two separate initiatives on the Grande Île (one carried out by Bioversity International, the other by the GIZ). The relevance of the choice was also confirmed a *posteriori*, insofar as the overall comparative approach has made it possible to expose the weaknesses of the Malagasy protocols on all issues related to territory, culture, and traditional institutions, and prompted our decision to make headway with these two cases.

What then governed the choice of survey methods was our desire to understand the BCPs in their development dynamics, seeking to shed light on their content through the negotiation and writing process. Two other factors were decisive. First, we considered that BCPs are also tools of mediation across scales and between different ontologies. Second, the number of instruments to be studied obliged us to increase the number of survey sites, while it is very difficult for us to bring together respondents over very long periods of time, in particular the farmers who, even if they are paid, are reluctant to devote several days to ethnographic work that takes them away from their agricultural activities.

For these reasons, we opted for two very short and rather original survey methods: the "group analysis" and what are called "ethnographic workshops". In Analavory, where they were first used, the two surveys were preceded by an exploratory survey which began with

a courtesy visit to the various regional, communal, and *fokontany* officials. Importantly, this enabled the identification of the contact persons with a view to the subsequent fieldwork: first of all, the active members of the FaMA cooperative and the leaders of the fokontany covered by the project (see above, Scheme 1) who had a good knowledge of the protocol development process in the area. The list was expanded to include the *ray aman-dreny* or *zokiolona* (village elders), who were approached mainly for their knowledge of community history, local customs, and seed exchange. Spread over a period of about two weeks (from 21 to 30 October 2019), this observation phase also allowed for a series of nondirective interviews in Malagasy, which were particularly valuable for refining the research protocol and placing the protocol in its social, economic, and environmental context (see Table A1).

The group analysis was conducted over two successive days (28 and 29 October 2019). It involved about fifteen participants and was mainly focused on the BCP. The main aim was to understand the dynamics of relationships (particularly within the community, but also between the community and the administration or the facilitator), the representations deployed, and the interests pursued around three themes: the origin of the protocol, the stages and process of developing and negotiating its content, and its impact on the functioning and cohesion of the community. Based on the model of Ricoeurian hermeneutics [43], articulating in a dialect way "explanation" and "understanding" [44], this methodology assumes, firstly, the immersion of the researcher in the research and their participation in the production of meaning by filling the hiatus between participants and researchers. Secondly, it holds the ambition and the virtue of encouraging the participants' reflexive skills: according to the model proposed by van Campenhoudt and colleagues [45], a sequence of analysis called "partial analysis" is based on a master narrative (chosen by the facilitator of our team) which gives rise to successive interpretations by the participants; the convergences and divergences are synthesised by the team of researchers, and then are the subject of proposals for further hypotheses and problematics. These are then submitted to the participants, who can thus directly compare their interpretation with that of the researchers. It is this dialectic between individual understanding, collective explanation (the researcher puts forward their theoretical knowledge), and then exchange of interpretations (we are back to understanding), which allows the reflexivity of the participants (and that of the researchers).

As for the "ethnographic workshop" (also called "intergenerational transmission" workshop), this was designed to question values, customs, and ways of life, so as to approach the world of the Analavory farmers. The method involved a dozen participants of different generations (four young people and five elders) who are placed in a context of intergenerational transmission of knowledge. One of the main interests of the method is to be able to capture social interactions, while observing the way in which knowledge is updated by young people in their daily lives. The method also encourages knowledge to be coconstructed between researchers and participants and combines several performative modes of expression—visual, theatrical, ritual, and artistic—with narration [46]. Three sessions, each lasting two hours, successively addressed three themes: (i) local customary practices, captured through a video sequence on the ceremony of turning of the bones ("double funerals") and the concept of *tanindrazana* (the land and soil of the ancestors); (ii) nature and the environment, also addressed through a video sequence on land issues and soil erosion; and (iii) seed exchanges within the community through the establishment of a participatory cartography of the fokontany covered by the BCP and an outline of the networks of seed exchange among the members of the cooperative.

Unfortunately, we have not yet been able to reproduce the same methods in the Mariarano and Betsako sites due to the current context of the pandemic. We thus drew on the research data produced by the action research carried out by one of our team members as part of the development of the protocol. In addition to the data collected during the entire action research process, which included both community workshops and community meetings, as well as multi-actor workshops involving representatives of the seven communities, the *Sojabe*, the regional managers of the forestry administration and private operators work-

ing in the area, and particularly those involved in the *Motrobe* value chain; our approach is also based on a systemic and critical review of the working documents and internal reports produced over the lifespan of the project. We also conducted semistructured interviews with stakeholders involved in the development of this specific protocol, as we did with the Analavory BCP (see Table A1).

## 3. Results

### 3.1. Development Process and Content of the Two BCPs

In this section, we examine the Mariarano and Betsako, and Analavory BCPs by critically shedding light on the development process, the objectives identified, and the role of each participant.

#### 3.1.1. Mariarano and Betsako BCP

Due to the island being a biodiversity hotspot with highly pronounced endemic wildlife, as well as the fact that the country's economy is largely agricultural and that it is one of the poorest countries in the world, Madagascar is an ideal experimental ground for new tools in terms of environmental management and biodiversity. These tools are politically Îlven and have been supported by international funders, including the World Bank, since the first national plan for the environment at the start of the 1990s [47]. Having ratified the CBD, Madagascar, alongside the other signatory states, focussed on the economic development of genetic resources as a method of funding and promoting the conservation of biodiversity [48]. This is apparent through the various instruments on the conservation of natural resources established since the start of the 1990s: management transfer contracts (e.g., the Integrated Conservation Development Projects (ICDP) approach, which allows people living in the vicinity of protected areas to benefit from cash income or small-scale agricultural and commercial development projects [49,50]), payments for environmental services, and mechanisms for reducing emissions from deforestation and the forest degradation (REDD+) [51,52]—in each case, the participative/community management of resources is supported by economic incentives, more broadly "institutional incentives", which are seen as crucial for changing individual behaviour [53] (para. 7), also given local communities' very low standard of living in Madagascar.

The emphasis on "incentives" should not come as a surprise, for they are at the heart of the CBD [54–59]. The "incentives" are primarily "economic". In terms of genetic resources or TK, it has been argued that "hardly any market value exists for the specific information they contain. This leads to a 'profitability gap' for the individual user of the biological resource, which can be defined as the difference in the profit between its sustainable and unsustainable utilization. In combining these conclusions with the outcome that the public value of biodiversity [and TK] cannot be covered by other mechanisms, it becomes clear that other mechanisms must be created" [60]. Among the economic incentives (such as taxes and subsidies), there are market-based incentives; for example, the creation of ownership rights over biodiversity and a market on which these rights can be traded. The CBD approach is, however, broader in scope. Building on Douglass North's (1990) work on institutional change, incentives are designed as a blend of "formal constraints" (economic and legal instruments, regulations, and public investment), "social constraints" (e.g., cultural norms, social conventions, mores, etiquette, traditions, and taboos), and "levels of compliance"—all of which make up what are called "institutional incentives". Therefore, incentives are not only so much about fine-tuned property rights over resources or knowledge or mechanisms of goods and services that flow from biological diversity, but rather about acting on local institutions and social norms in order to reinforce enabling social constraints and drive necessary changes [60,61].

The BCPs are the latest conservation tools that have been experimented with in the Grande Île and are thus being deployed in a context where the stakeholders, notably funders or project leaders, appear to be completely immersed in a philosophy that can be qualified as economist and neo-institutionalist, resolutely focussed on institutional control

over the management of nature by local communities. This influence can clearly be seen in the Malagasy BCPs, which are constructed as economic and instrumental mechanisms.

In Mariarano and Betsako, this is even clearer, given that the BCP of the seven communities was established on the basis of the existence of management transfer contracts [62] in these areas. These contracts, it should be recalled, are tripartite agreements between the regional departments of the Ministry of the Environment and Sustainable Development, the decentralised regional authorities (the communes), and the local communities (referred to in Malagasy as *Vondron'Olona Ifotony*, VOI). As local structures forming a "voluntary group of individuals united by the same interests and obeying the rules governing the life of the commune, known as the *Dina*" (Art. 3 of the Gelose law), the VOI were clearly a magnet for the BCP. This is understandable insofar as the VOIs, which were favoured by the Ministry of the Environment, already offered an operational and well-known tool. Depending on location, the VOIs bring together the residents of a hamlet, a village, or group of villages. Above all, they enjoy legal status in private law, on an associative basis [63]. In contrast to an NGO, however, the VOIs can have a profit-making objective and can conduct income-generating activities [64]. These activities must respond to the two-fold objective of the sustainable development and conservation of natural resources, which is formalised through the VOI Management and Organisation Plan and the Communal Development Plans for the communes [23]. In Mariarano and Betsako, this takes the form of the exploitation and improvement of local economic industries, including that of *Motrobe,* which, through the substitution effect, encourages local populations to take part in activities that develop rather than damage the environment [65]. The attraction of management transfer in the BCPs is such that it can be questioned whether the value chain in question really falls within the rules of ABS as set by Decree 2017–066 of 31 January 2017 on the regulation of access and benefit sharing arising from the use of genetic resources. Indeed, from the bioprospectors' point of view, it does not appear to use the plant for its "genetic resources" as part of research and development (R&D) activities, but only to produce essential oils (in other words, only biological resources are being sought after). Like the Nagoya Protocol itself (Art. 2(c)), the Malagasy Decree covers the "utilization of genetic resources", i.e., "research and development on the genetic and/or biochemical composition of genetic resources, including through the application of biotechnology". The real question is whether bioprospectors only produce essential oils with the plant or whether the biochemical composition of the plant is also the subject of a R&D process [14].

Of the seven sites involved in the BCP, six already had a VOI in place (Komamy, Mariarano, Tanandava, Marosakoa, Tsiankira, and Ankilahila) and the other (Tsiankombezo) is in the process of requesting a management transfer agreement from the decentralised authorities. In each of these sites, economic activities around the *Motrobe* plant were already in place before the project began. Some VOIs had already built economic partnerships with private companies like Aroma Forest [66], Homeopharma, and smaller collectors–exporters such as the companies Faly and Dety Body Nature [67], thus establishing the first value chains. The *Motrobe* value chain was selected from a preliminary study of the forestry value chains carried out in 2015 (the two other possibilities were *Katrafay (Cedrelopsis grevei)* and *Aloe macroclada* [68]) because it appeared to GIZ that it was possible to identify a large number of the participants at each link in the value chain and because it was well structured.

The choice of an existing, well-structured sector is understandable from the point of view of the experimental approach and the "feedback loop" mechanism [69] chosen by the project sponsors. Developed in the context of reflections that were underway on the national ABS legal framework, the Mariarano and Betsako BCP was above all a pilot project aiming to feed into work on legislation and regulations, while ensuring local validation of the normative choices that had been made on the national level. It can also be considered as a pilot project insofar as, for its developers, in particular GIZ and Natural Justice, it consisted of being able to measure how the local communities (the *fokonolona* as a whole, not only the VOIs) would react to the new instrument when it was put into practice as

part of a value chain that affected their everyday life. The situation on site also offered the unique opportunity to be able to evaluate the impact of the BCP on issues of prior consent and benefit sharing within intracommunity relationships, as well as with economic operators.

The BCP was rolled out at a difficult time, where disputes around the *Motrobe* value chain were mounting. First, the procedures for issuing permits to collect forest products provoked social tensions, insofar as the various people involved considered them to be unclear [70,71]. In the intracommunity relationships, opposition between members of the VOIs, the *fokontany* (the smallest administrative unit in Madagascar) leaders, and the rest of the community or *fokonolona* was criticised, with the rest of the community accusing the VOIs in particular of appropriating all the benefits of the exploitation, by excluding the relevant administrative leaders on the local level—the heads of the *fokontany* and communal leaders—but also the existing traditional authorities from the process of issuing the permits. In Mariarano, for example, some actors said "it appears that the forest only belongs to the VOIs" ("*tsy hoe lasa fananan'ny VOI ny ala*"). Some members of the community also complained that the *Sojabe*, the village elders who continue to exercise their customary powers in the social life of the *fokonolona*, had been completely marginalised. For their part, the local communities criticised the private operators for profiting from their weakness and illiteracy, by imposing practices such as the use of collection permits which had expired or the use of permits obtained from the central ministry [72,73] and hence with little regard for the regional and local authorities. For example, in Mariarano and Betsako, the interviews carried out with the technical manager of PAGE/GIZ allowed us to reveal the existence of a US NGO which had been working since 2009 in the forests managed by the VOI in Mariarano and whose research permit in the village was issued directly by the central ministry, thus short-circuiting the regional Directorate in Majunga.

It should also be noted that the collection permits reawakened distrust among the local community with regard to the state, relationships which date back to the colonial period and in particular the practice of various "*hetra*" or taxes that were taken by the colonial administration [74,75]. Finally, in the exploitation of the *Motrobe* value chain, the communities remained prisoners of an oligopolistic market due to the geographic distance and enclosed nature of the sites: a market which was therefore dominated by a few private operators setting very low prices per kilo of leaves collected (200 Ar/kg). The activity was, therefore, far from being profitable, because the local communities had to invest long hours walking through the forest for a paltry collection of 10 kg per person per day (e.g., a working day as an agricultural worker pays 3000 Ar/day, clearly more profitable than a day collecting *Motrobe* which pays 2000 Ar/day). This is, however, a key activity which part of the community is forced to undertake to increase their farming income, particularly during the off-season—generally from January to May in the Boeny Region, a period which coincides with a drop in precipitation and the waiting period for the first rice harvests [76].

The conceptual closure of the BCP around the value chain, thus to the detriment of issues relating to tradition, land, and bioculturality—which is only briefly alluded to in the title ("BCP of local communities custodians of biodiversity and holders of traditional knowledge of communities") and through the incidental reference to "territories of life" in "living area" (a reference clearly imprinted into the ICCA "Indigenous and community areas and territories" Consortium—see below pp. 20–21)—is first and foremost provoked by the search for a site in which local procedures around free, prior, and informed consent (FPIC) and new rules around benefit sharing can be experimented with (the very rationale behind the "pilot" project). At the same time, the potential value of activities of bioprospection and biotrade for rural development is not unfamiliar to the approach, and it is clear that the ABS Capacity Development Initiative (known in short as the ABS Initiative), a multistakeholder initiative hosted by the German Federal Ministry for Economic Cooperation and Development (BMZ) and administered by GIZ, which is associated with the UEBT (Union for Ethical BioTrade), has made the development of "ABS-compliant biotrade value chains" one of the spearheads of its work. As representatives of Natural Justice and

PAGE/GIZ stated during a recent webinar, the "organisation of the local community for dialogue and negotiation with external entities, particularly the private sector, regarding the development of natural resources" has always been borne in mind (Webinar entitled "Biocultural Community Protocols in Madagascar: Sharing experiences and lessons learned in the context of the Access and Benefit-Sharing from the Promotion of Biodiversity and Related Traditional Knowledge" on 23/06/2020). However, the more immediate objective emerges when the BCP encounters the coordination difficulties with which the VOI (and more broadly the *fokonolona*) are confronted on a daily basis, whether this be intracommunal relationships or relationships with private operators and the forestry authorities.

As a result, the Mariarano and Betsako BCP remains blocked by the issue of the *Motrobe* plant and the management transfer, and almost the entire document can be read as a short guide to strengthening the capacity of local actors and improving collective action. Other than paragraph V ("Local community engagement for the sustainability and use of biodiversity"), which establishes the rules around collective choice (the period and quantity of the harvest and harvest methods) and which outlines a monitoring mechanism, and paragraph VIII ("Conflict resolution"), the entire protocol addresses coordination of the decision-making process between *fokontany* and local communities, and the sharing of benefits arising from harvest activities (respectively paragraphs II ("Method for granting prior consent for the issuance of a permit") and VII ("Benefit sharing method"), which are the most developed) [77] (see Figure A1). This is so true that if we exclude the issue of the *Motrobe* plant, which is never mentioned but appears discretely in a few photographs, it is difficult to understand the institutional organisation that has been chosen.

Reading between the lines of the Analavory BCP, it also becomes apparent that the BCP is reduced to a simple instrument for institutional organisation and incentives. The context is, nonetheless, quite different, insofar as the Mariarano and Betsako BCP was principally focussed on issues of ABS around the *Motrobe* value chain while that of Analavory was first established to enable agricultural seed exchanges—plant genetic resources for food and agriculture of the ITPGRFA—before returning to the scope of the Nagoya Protocol at the end of a tortuous process of development.

### 3.1.2. Analavory Farmers' BCP

It should be recalled that the particularity of the Analavory farmers' BCP compared to the Mariarano and Betsako BCP is that it was developed as part of a mutually supported implementation of the Nagoya Protocol and the ITPGRFA. Funded by the Darwin Initiative [78] and taking place between April 2015 and March 2018, the project was led by the research for development organisation, Bioversity International. Its primary aim was to simultaneously develop in Madagascar and Benin a legal framework ensuring the connection of the two main international legal regimes for the governance of biodiversity (CBD/Nagoya Protocol and ITPGRFA), while ensuring greater protection for local communities [79]. In this regard, it was crucial for the project holder that certain communities were chosen in each country, not to act as potential "suppliers" of genetic resources, but as "good candidates for receiving genetic resources on the basis that they need [ . . . ] adapted germplasm to respond to climate changes or soil degradation [ . . . ]" [79]. One of the two Malagasy communities (the other community is that of Ampangalantsary located in the east of the island [80]) had, therefore, to be integrated into the project for its agricultural activity—and not for the potential wealth of its genetic resources likely to be transferred through ABS agreements—and its germplasm needs (PGRFA) likely to be met through the multilateral system (MLS) of the ITPGRFA. The rural community of Analavory met these criteria both because of its agricultural activity and the lack of genetic resources of interest (notably of agricultural interest) in the area. It is worth noting here that if Madagascar is certainly an extremely diverse country, it is not a centre of origin of cultivated plants [81]. In terms of the main plants cultivated, Madagascar's degree of dependence with regard to crop genetic resources coming from the primary regions of agricultural plant biodiversity is very high (between 94% and 100%) [82].

When the process of developing the BCP began, a variety of objectives and points of view emerged from the actors involved, leading to tensions between the project holder and Natural Justice, the subcontractor. For Bioversity International, according to a relatively dated framework within the FAO's Commission on Genetic Resources [83], PGRFAs have the economic attributes of an "impure public good" [30,84]. Their attributes expose them to "under-use" rather than to "over-use" [85–87]. Furthermore, while the use and exchange of these resources produce positive externalities in the form of "use value" or "option value", the farmers at the local level appropriate none of these values. The entire issue consists, therefore, of rolling out "incentives" in such a way as to ensure that the farmers do not abandon cultural or traditional varieties for potentially more lucrative activities [88]. The ABS mechanisms may play the role of "market-based incentives" by assigning a value to the PGRFA, while simultaneously ensuring the economic development of communities. Given the low value of PGRFA in Analavory, the approach was slightly different, but did fall within the more general discussion on "institutional incentives" mentioned above: this consists of resolving problems around collective action, through a battery of measures aiming to improve farmers' capacity to identify the plant genetic resources they need (which involves assessing the state of genetic resources in their farms), obtaining them from sources external to the community, testing and multiplying these seeds, and adapting the material through participatory plant breeding [30]. The issue was, therefore, still to avoid "under-use", but under-use was not considered as having been provoked by the lack of economic incentives; the main problem is linked to institutional and capacity weaknesses that prevent farmers from taking "advantage of the technological and organizational developments that have changed the shape and functioning of the global crop commons over recent decades" [30]. In other words, from the point of view of the "global crop commons" that the ITPGRFA and the MLS represent, smallholder farmers do not play a full role in the conservation, use, and improvement of crop genetic diversity if they are unable to access the largest pool of germplasm available.

Such an approach had little chance of aligning with that advocated by Natural Justice around BCPs for two reasons: first the BCPs are first and foremost linked to the Nagoya Protocol and ABS, and the connection to the ITPGRFAs is unprecedented. Secondly, and above all, through culture and experience, Natural Justice has developed BCPs as tools for protecting the biocultural heritage of communities within the context of bioprospection activities.

These two broadly opposing approaches created recurring tensions between the project holder and the facilitator, whether this be around discussions on the schedule, additional funds sought by Natural Justice for the development of the protocol and the seed bank, or disagreements as to the publication of the BCP itself.

Constrained by time and project funding, Natural Justice complained about having to negotiate a protocol quickly that should have been ready at the end of the project, in order to be able to conduct a series of germplasm exchanges with research centres and between communities involved in the project. It also criticised an objective that had not been discussed with the communities.

*"Umm, and that's something that was a bit* [silence] *difficult for us as Natural Justice, which facilitated the entire process, and I think it's something that . . . that . . . that we will . . . umm, really integrate into the overall guidance on community protocols, because we are going to do that. It is this relationship, well, this . . . umm . . . mediation between the local communities on one hand and those that finance the entire project on the other. Because the bodies that are involved, such as the Darwin project which, they have got it into their heads that 'hey, we'll develop a community protocol for this purpose!' . . . for example* [continuous laughing] *. . . and on the other hand, there are the communities who have their own priority as well which might be different to . . . umm . . . those of . . . the . . . funders, technical and financial partners. So it's really there that we try* [word missing] *the two, the . . . two parties by establishing ourselves as a mediator. For example, if you look at the reports as well, if we had to . . . well add to the Darwin budget*

*. . . the Darwin project it's because of the community protocols* [laughs], *because I was stubborn in fact, I thought that X* [the project holder for Bioversity International] *blamed me a bit, and the Ministry blamed me a bit saying: 'Well, listen guys, you're not giving me enough time, for me or for the farmers!'—as if the community protocol had to be finished* [continuous laughing] *by the end of . . . umm . . . the project* [laughs] *to be used for that purpose."*

(Interview, Natural Justice, facilitator, 05/05/2019)

In terms of the project holder, there was an acute awareness of the facilitator's progressive transformation of the main objective of the protocol, a transformation to which Biodiversity International ended up agreeing, both in terms of agreeing to the timescale and additional funding to continue the work around ABS and bioprospection, as well as the establishment of a seed bank. However, the holder expressed scepticism, as there was little doubt in their mind that a BCP constructed around ABS and bioprospection does not make much sense for a farming community such as Analavory:

*"I mean the other problem with all these protocols from the Nagoya point of view is . . . umm . . . you can build a fence around all kinds of things and that doesn't create a demand for it and that doesn't make it useful* [laughs]*. [ . . . ] You know where in communities where it's purely agricultural they're not . . . they're working on crops, from other centres of origin and . . . you know . . . . put a lot of effort into developing a protocol but . . . if there's no interest in getting stuff from them, you never get . . . you know . . . it's another thing."*

(Interview, Bioversity International, project sponsor, 23/05/2019)

At this stage, the complexity of the task of facilitator becomes clearer, torn as they are between the objective set by the project sponsor and which obliges them to skew the BCP towards the ITPGRFA multilateral system, and their desire to introduce the issue of biocultural heritage, but which in turn requires the BCP to be reoriented towards the Nagoya Protocol. The problem is that they are immediately caught out by the realities on the ground, which confront them with a great poverty of genetic resources, thus compromising the founding of the protocol in ABS and bioprospection. This fragility is further confirmed by the current situation in the area, where it is obvious that the BCP is not used due to a lack of genetic resources—with the exception of a few endemic plants used in traditional pharmacopoeia [89]—liable to be covered by ABS agreements with bioprospectors.

This blurring of objectives is reflected in the content of the BCP itself, which has certainly been reoriented towards the ABS mechanisms as, of the 27 pages that make it up, only three are devoted to PGRFA and the MSL. And there is little in the rest of the document which addresses genetic resources "other than" the PGRFA, concerns which are traditionally associated with the defence of biocultural heritage (see Figure A2). There are two summary pages that address TK, but in contrast there is nothing on land rights or cultural rights, no mention of the role of "stewards" that the local community would play, and very limited mention of customary rights (on the BCP content, see [90]).

There was, however, the central issue of the "ontological" value [91] that seeds had for the Analavory farmers, in connection with the extremely strong customary rules for the group's identity. However, this remained unexplored and was only revealed by our ethnographical workshop, which recognised its ongoing sensitive: *"Sakafo ve dia asiana contrat*?"—"Do we really need to have a contract for food?" The combination of the two terms, "*contract*" and "*sakafo*", i.e., food—in this case, seeds—is incongruous for the farmers.

As the Malagasy people say: "*sakafo masaka, tsy mba manan-tompo*"—"prepared food has no owner"; this is about sharing. This falls within a system of ideas, which are based around a hub of tomb-filiation-ancestry, the tomb being, as Bloch clearly demonstrated, "the ultimate criteria of membership" [92]. It is within this hub that identity is formed, which continues to be linked to the *tanindrazana* (the land and soil of the ancestors) where the family tomb is located. The importance of this is particularly clear, especially in the study region, in the ceremony of turning of the bones: the *famadihana* ("double funerals"). In the

cosmogonic system in Madagascar, the ritual of transferring the bones into the new tomb or changing the ancestor's shroud expresses the reciprocal and permanent transactions which link the living and the ancestors (who are never really "dead"). The expensive ceremony—which involves the entire productive system well in advance—is first and foremost an obligation resulting from the first legacy of the deceased. The legacy (*lova*) comes with responsibilities in perpetuity, such as maintaining the tomb and the *famadihana*. A range of obligations including to plant the "*angady*" (a traditional spade) in the ancestral rice fields to ensure the food security of the family and, above all, to rapidly collect wealth to worship the ancestor [93]. However, the ritual never wipes out the debt, because the descendants are not satisfied simply with giving it back—they have to "re-give", so to speak, by calling for a new gift and then tracing a never-ending circle—which ensures the continuity of the vital flow or *aina*, focussed on the *razana* (the ancestor) [93]. "*Raha razana tsy hitahy fohazo hiady voamanga*" ("When an ancestor doesn't watch over us, wake them up and send them to pick potatoes") goes a Malagasy saying: the ancestor is not *dead*—i.e., is only of use—if they serve their descendants.

The importance of the "growth process" which, as Eva Keller recalls, is the first criterion of success for the Malagasy farmer, can thus be understood [94]. Farm, cultivate, grow—in order to hope to be able to fastidiously honour the *razana* and thus capture additional vitality – but which can only be achieved by following the "normal course of things" which is set by God (*Zanahary*), i.e., by scrupulously and manifestly respecting the intermediate divinities (the "*lolon-kazo*" or the "tree spirit", the Chthonian powers, etc.) and the *razana* themselves [95]. This is the subject of offerings, prayers, and even *ody andro* or *ody avandra*, i.e., spells against (poor) weather ("*andro*") or hail ("*avandra*") which aim to ward off climatic risks. In Analavory and elsewhere in Madagascar, the spell-maker can converse with the beyond, with ancestors, in such a way as to ensure things run their usual course. Ultimately, the farmer must respect the *fadin-tany*, i.e., taboos regarding the land [93]. It is thus *fady* (forbidden) in Analavory to take green wood, peanuts, or Bambara nuts into the village during the rainy season, as they will attract lightning. For the same reasons, it is also *fady* to cut stones during the rainy season. Farmers therefore operate within a dense normative network, which has two poles, the "*tsiny*" and the "*tody*". *Tsiny* or "blame" is the "readiness to answer" [96], in other words the acceptance by the individual that they will have to answer for any behaviour that may be contrary to custom. *Tody*, on the other hand, is the "arrival" or "return" of things that the world order imposes on human action. It is an impersonal and automatic sanction attached to the act itself, but the timeframe (near or far future) cannot be anticipated [97].

In such a system, which constantly weaves together agricultural production and the creation of a "social symbolic" [93] system, some exchanges necessarily bring into question the continuity of the group and constitute support for identity [98]. They respond to their own logic, as is the case of seeds-food. When foodstuffs are shared (raw or prepared), the Analavory farmers say this means that there is trust among the guests. If a "contract" has to be signed, this means that the trust is in doubt, trust has not been acquired—which changes the status of the food (the "*safako*" is also a vector for poison or witchcraft "*vorika*") [99]. Any negotiation or commitment (*fifanekena, fifampiraharahana, fifandaminana,* or *fifanarahana*) takes place orally. No written contract is prepared. Belief in the "*tsiny*" and "*tody*" is enough to guarantee that each party respects the commitments. In contrast, written agreements represent the state/the *fanjakana*, which requires signatures and leads to distrust. Seeds-foods are thus, in this context, either redistributed in small quantities to close relatives (e.g., a young couple) or exchanged against seeds of another variety. They do not follow the trajectory of technical seeds (modern cultivars), which are usually the only ones to be the subject of a commercial exchange. Technical seeds have been significantly successful on site, but the farmers reserve a special status for them, due to their origin, their mode of circulation, their destination, and their temporality—they follow "trends" ("fashions") and are never fixed.

This last example revealed the feature that the Analavory protocol was lacking, and which prevented it from being a real tool for protecting communities, their worldview, and their heritage. It also highlighted the factors that may impact upon the development process and compromise better consideration of the biocultural heritage of communities. In order to examine these two questions in more depth, we must first look at the emergence of the biocultural approach in conservation policies, particularly in Madagascar, keeping sight of the need to further examine the "biocultural gap" in the two Malagasy BCPs.

### 3.2. Importance of Taking the Biocultural Dimension into Account

IPLCs have been increasingly described as playing an important role in the sustainable management of complex ecological systems since the late 1980s [100]. The Belém Declaration (1988) was a watershed moment in the international promotion of the idea that indigenous peoples are the "custodians" of the planet's genetic resources [101], demonstrating the "inextricable link between cultural and biological diversity" [102,103]. For Maffi [104], biocultural diversity "comprises the diversity of life in all of its manifestations—biological, cultural, and linguistic—which are interrelated (and likely co-evolved) within a complex socio-ecological adaptive system" [105].

To explain the growing attention paid to this concept, the three major movements [18,106,107] at the heart of the development of the biocultural nexus should be briefly recalled, as they shed light upon the concept and bring to light any ambiguities. It is by analysing each of these movements and the way in which they influenced conservation policies in Madagascar (Section 3.2.1) that we can better measure the "biodiversity gap" in the Malagasy BCPs (Section 3.2.2).

3.2.1. Movements at the Heart of the Emergence of the Concept of Biocultural Diversity and Their Influence on Conservation Policies in Madagascar

The first movement is referred to by the term "fortress conservation" [108]. Directly linked to the preservation of nature, it advocates the sanctuarisation of natural spaces, the establishment of protected areas to the detriment of IPLCs. In Madagascar, this movement was characteristic of the colonial period, during which the colonial administrators unceremoniously decried the threat that the local communities and farmers posed to the island's majestic forests [109]. This political embodiment of this movement was as simple as its effect was radical and fatal: protected areas were to be promoted and placed outside any human influence [110]. Historically, the creation of protected areas in Madagascar can be broken down into three phases [111]: (i) the creation of reserves during the colonial period (1896–1960) [110,112]; (ii) the expansion of national parks in 1990–2003 during the implementation of the National Environmental Action Plan [112]; and (iii) the expansion of protected areas with different models and categories of governance from 2004 as part of the Durban Vision [113]. These three phases saw the scale of the protected areas in Madagascar increase from 450,000 hectares in the 1980s [114] to nearly six million hectares (10% of the island's surface area) in 2010.

The second movement is that of "development". The accent is upon "ordinary people" and the emergence of a new conceptual language around "capacity building, grassroots participation, decentralization and sound environmental practices" [115], which must not conceal an approach that places responsibility for environmental damage on the poor [116,117]. The growing critique of the first model in the 1970s, as well as the recognition of the relevance of the notion of culture, which had previously been overlooked, gave rise to this second movement. It saw the rise of a new concept: IPLCs are depositories of knowledge beneficial for the conservation of the environment [118]. In Madagascar, this model was promoted in the 1990s as part of the Washington Consensus [119]. It was accompanied by a shift in the way in which local communities and farmers were represented, no longer being the "enemies of nature" but as "premodern" and caught up in the "spiral of vicious circles": strong demographic growth that generated poverty, which in turn caused the deterioration of natural resources [120–123]. For the World Bank experts who contributed to the problematisation of the Malagasy environmental issue, part

of the solution therefore resides in the fight against rural poverty in all its forms (land security, agricultural intensification, income-generating activities, etc.) [117]. Also, under the influence of major players in global environmental governance (UNPD, WCS, USAID, etc.), Madagascar established a National Environmental Action Plan—divided into three 5-year phases, each in the form of an Environmental Programme (EP1, EP2, and EP3). Under the EP1 (1990–1995) [122], Madagascar committed to expanding protected areas by creating a national network of protected areas; the ANGAP (or National Association for the Management of Protected Areas which became Madagascar National Park from 2007) was created to manage protected areas [50]. However, the evaluation of the first programme revealed mixed results. Considered by some even as an "impoverishing" programme [122], the "sanctuarisation" model of the EP1 was accused of being at the origin of "famine conditions" of local populations excluded from their natural spaces [122], and that despite the integrated conservation and development projects (ICDP) implemented around the protected areas which were supposed to improve their standard of living [50]. More broadly, it was the over centralisation of the management of natural resources that was criticised [124].

The third movement, which also contributed to making the activity of local populations visible, was the movement in favour of the indigenous peoples, which in this model is inseparable from the history of biocultural diversity [106]. This movement took root in the 1960s, with the emergence of a new generation of young indigenous people, refined connoisseurs of the dominant legal system, who were able to hold forth on the "survival" of indigenous people as "distinct communities with historically based cultures, political institutions, and entitlements to land" [125]. Then, in the 1970s, there was increasingly active participation by representatives of indigenous peoples in major international conferences, followed a little later by academics and NGOs. In 1980, a considerable leap forward was made with the adoption and entry into force of the first major international instrument, ILO Convention No. 169 on Indigenous and Tribal Peoples (1989) [126], which was extended almost two decades later by the United Nations Declaration on the Rights of Indigenous Peoples (UNDRIP) [127]. What has to be carefully noted is that "cultural diversity and environmental conservation were crucial issues in the arguments about 'indigenous peoples' and their rights to lands, local resources, self-determination, and particular identities from the beginning. A particular relationship to the places they inhabit, often related to historical continuity, is at the core of their claims to lands and territories and discussed in the context of particular conceptualizations of and relations to 'nature' different from 'modern' environmental relations" [106,118].

This last movement arrived late in Madagascar, in any case after Resolution 61/295 of the United Nations General Assembly (17 September 2007) which adopted the UNDRIP (A/RES/61/295), insofar as Madagascar had not ratified ILO Convention No. 169. It was also at the end of the 2000s that the voices of civil society organisations supporting the ground-up governance of natural resources (e.g., Tafo Mihaavo, which is the national network of local communities (*fokonolona*) managing natural resources [128]) and equality of land rights (e.g., the Plateforme Solidarité des Intervenants sur le Foncier, SIF) and many other bodies began to increase.

It was in this crucible of contradictory movements, which continue to circulate in Madagascar, that the idea began to progressively form of an inextricable link between humans and the environment, and which managed to formalise the concept of "biocultural diversity", the presence of which has continued to be visible in the past five years. The context is important here: as evidenced by the associated concept of "steward of biodiversity" [129], "biocultural diversity"—regardless of its scientific validity—has a strong political dimension which echoes the initiatives at the end of the 1980s [101] aiming to protect the rights of indigenous peoples and rural communities regarding their land and cultural heritage [130,131].

In Madagascar, the influence of "cultural biodiversity" is difficult to discern in the country's policies and legislation. Decree No. 2017-066 of 31 January 2017 regarding

regulation of access and benefit sharing arising from the use of genetic resources alludes to BCPs but in the form of a "tool developed by the communities" to document "traditional values and practices" (Art. 14). Meanwhile, the preliminary draft of the Interministerial Order on implementing rules for Decree No. 2017-066 contains mentions of the "Biocultural Community Protocols", which are taken as examples of "tools" that could allow for the negotiation of Mutually Agreed Terms (MAT) (Art. 22).

The Gelose Law was also innovative in that it allowed for the consideration, for the first time, of traditional rights and local communities. Under the impetus of international environmental actors and brokers (WWF, UNDP, etc.), Madagascar soon ratified the Convention on Biological Diversity, signed in Rio in 1992 (Law No. 95-013 of 9 August 1995). The global environmental discourse (e.g., the Brundtland Report (1987) and the Declaration of Belém (1988) mentioned above), which now referred to the local communities as "custodians" or "stewards" of nature [107], inevitably accelerated the consideration of an "integrated and participative" dimension involving rural populations. While the conviction that the success of any conservation work and forest management involves an "arrangement of the land <*terroir*> as a whole" [49] took hold on the ground, including local communities and their means of survival, a U-turn towards a new policy combining the sustainable management of forests and the decentralisation of governance began to be drawn up in the context of the EP2 (1996–2002). Two texts are emblematic: the Law 1996-025 on the secure local management of natural resources (the Gelose Act) and Law No. 1997-017 on the revision of the forestry legislation. The main outcome of these two laws was the delegation or transfer of the management of natural resources to local populations (also referred to as secure local management or Gelose). In 2017, nearly 1248 transfer management agreements were signed throughout the whole island [132].

However, one of the main criticisms of the Gelose Act is that it poses a certain number of problems regarding the adoption and real application of the *dina*, given the distinction between the "endogenous" *dina* (which express customary laws) and the "Gelose" *dina* (which integrate modern law) [63]. However, the biggest criticism was the fact that the members of the VOI were only one part of the entire community (*fokonolona*)—consisting of those members of the community who voluntarily join the VOI [133]—while the VOI were the only legal structure recognised by the state to manage the resources. Moreover, the administrative limits imposed through the subdivisions into *fokontany* in the 1970s and recognised by the law in force only rarely took into account the real land boundaries that already existed and which continued to be used by the *fokonolona*. It thus became obvious that the VOI as well as the Gelose *dina* had little legitimacy in the eyes of the *fokonolona*.

These observations enabled environmental researchers and national and foreign activists working in the country to pay greater attention to the need to focus development and conservation of resources on the *fokonolona*. The creation of Tafo Mihaavo, in 2012, also contributed towards this dynamic. In its "Declaration of Anja from the general assembly of the local communities managing natural resources for an efficient governance and sustainable management of natural resources based on the values of the *Fokonolona*" (2012), Tafo Mihaavo clearly stated that "the *Fokonolona*, the social structure based on Malagasy values, are the most stable, well-established and durable structure that the Malagasy communities accept, especially when the country undergoes a crisis. Neither colonization nor the successive regimes could erase it [ . . . ]" [134]. Its Declaration of Mantasoa (2015) added that "The *Fokonolona* is a community of *Olombelona*: Everyone living in the same area <*terroir*> and aspiring that the common good (cultural identity and natural wealth) is maintained and secured in the very long term, and to remain the owner and once again the governor of their natural and cultural environment through collective agreements (*Dina*)" [135,136].

Since its creation in 2012, Tafo Mihaavo has joined the ICCA Consortium created in 2010 during the fourth IUCN congress in Barcelona, an international movement promoting equity in conservation, fighting for self-determination and sustainable living for indigenous peoples and local communities [137].

ICCAs are "natural and/or modified ecosystems, containing significant biodiversity values, ecological benefits and cultural values, voluntarily conserved by indigenous peoples and local communities, through customary laws or other effective means" [138].

Referred to as "communities which are custodians of ICCAs" [139] in the country, the communities which are members of Tafo Mihaavo were able to identify and improve the protection of 14 ICCAs spread across five regions of the island, with the financial support of the UNDP Global Environment Facility (GEF) Small Grants Programme (SGP) [140]. It should also be noted that Tafo Mihaavo, with its various supporting partners within the NGO Fanonga, campaigns for an update to Orders Nos. 73-009 of 19 March 1973 and 73-010 of 24 March 1973 on the rights and responsibilities of *fokonolona*, with a view to better taking into account the role of the *fokonolona* in the management of common goods, in particular natural resources [133,135,141].

3.2.2. The "Biocultural" Gap in the Two Malagasy Community Protocols

Although it is difficult to precisely trace the origin of BCPs, understanding their genesis and the role that their creators and promotors assigned to them enables us to have a better grasp of the gaps that we have identified in the two Mariarano and Betsako BCPs.

- A reminder of the "biocultural" aspect of community protocols

BCPs appear to have been inspired by "nonlegal" instruments developed in Australia and New Zealand for the protection of the intangible heritage of indigenous peoples. Kelly Bannister recently listed many of these under the name of "Indigenous Statements and Declarations" and "Community Research Protocols" [142]. Although each of these categories includes quite different tools, one thing they undoubtedly have in common is the fact that they are "community-level instruments", i.e., instruments that are produced by the communities themselves [142].

Simultaneously, while IPLCs seek to consolidate their position through environmental law, initiatives are multiplying around "Indigenous peoples' declarations and statements on equitable research relationships", "Community research agreements", "Community protocols", and "Community codes of conduct" which aim to define the terms of new relationships between IPLCs on the one hand, and researchers and private companies on the other, in the field of biodiversity [143].

Among these initiatives, those around BCPs have gained in popularity due to the considerable publicity given to the Intercommunity Benefit Sharing Agreement in the Parque de la Papa, in Pisac (in the Cuzco region of Peru) [144], before being promoted internationally by Kabir Sanjay Bavikatte and Harry Jonas [18,19,29,145,146], founders of the NGO Natural Justice, who worked alongside the African Group of States, to introduce them into the Nagoya Protocol in 2010.

The work of these two lawyers (and their collaborators), as well as the BCPs that Natural Justice was able to rapidly develop over the course of the past 10 years around the world [147–149], clearly shows that there is a close link between the "cultural protocols" mentioned above and BCPs. The bedrock—perhaps less visible in "cultural protocols" [142,150]—is customary law and local institutions, objectives and development priorities of the people and communities, their holistic way of life, and their connections to the land. However, the BCPs also visibly change the political scope of the protocols by setting them within an international discourse that refers to IPLCs as "stewards" of the environment, which obviously raises the problematic myth of the "good ecological savage" once again [151,152], but which cannot be separated from a form of "strategic essentialism" [153], that has paid off [154] (e.g., "traditional stewardship" is now enshrined in the Tkarihawaié:ri Code of Ethical Conduct to Ensure Respect for the Cultural and Intellectual Heritage of Indigenous and Local Communities [155]) especially at a specific moment in time when non-naturalist ontologies appear to be inspiring fairly significant legal changes around the world [156].

The BCPs primarily promote the "ethic of stewardship" [19,155,157–160], which is claimed to describe a holistic way of life, rooted in the land and territory. In their theo-

retical formulation—but which can also be seen practically at work in a certain number of BCPs developed in Latin America (see, *inter alia*, the BCP of Capulálpam de Méndez, Oaxaca [161]; Ek Balam BCP [162]; BCP for Cerrado Raizeiras: the customary right of healers in the Cerrado biome of Brazil [163])—the BCPs strive to ensure the "recognition and respect for cultural knowledge, language, spirituality and all aspects of traditional lifeways, and protection of these from exploitation [ . . . ]" [164]. Much like Posey, who had already formulated the idea of a basket of rights ("traditional resources rights") [165], i.e., an "integrated rights approach" [166] to support the holistic way of life and identity of indigenous peoples, Kabir Bavikatte and Natural Justice always strove, through the BCPs, to reconstitute "the total mosaic of a community life that is fragmented under different laws and policies, with the understanding that the conservation of Nature is a result of a holistic way of life" [18].

In the context of the Nagoya Protocol, BCPs are often reduced to technical tools that enable at least the partial codification [16] of "protocols" or "procedures"—often presented as ancient, traditional, or long-standing—through which the IPLCs managed their resources and knowledge and defined the conditions for their use and sharing within the community or even beyond it [142]. Resituated in a broader political context, which is marked by a singular redevelopment of TK in the fields of ecology and conservation (see, e.g., IPEBS report on its Seventh Session [167] or the broad definition of protocols in the Mo'otz Kuxtal voluntary guidelines [15]), as well as the modes of life and non-naturalist worldviews of environmental ethics, BCPs had a political reach that should not be under-estimated [90,168].

Thus, according to their proponents, BCPs thus offer a unique opportunity to communities to self-define, to reaffirm their rights over their territories, resources, and associated knowledge, while highlighting their values and worldviews. These distinctive traits, which characterise the biocultural approach to community protocols, were particularly lacking in the Malagasy protocols we examined.

- The absence of the "biocultural" in the Malagasy community protocols

Examination of the two Malagasy BCPs struggles to identify a real investment in the issues relating to the stewardship of nature, of biodiversity, the ethic of stewardship, the links between way of life, traditional institutions, customary law, and local knowledge. The two documents are almost exclusively constructed around the issues of free, informed, and prior consent, and benefit sharing [169].

The lack of the "biocultural" dimension is particularly visible with regard to what can be qualified as the "biocultural hub" [106], namely the land—an unbreakable and unique hub which, enriched with collective practices and knowledge, defines (and redefines) a whole range of relationships [170] which go beyond human relationships alone and create the basis of the group's responsibility [107]. The "lands, territories and resources of indigenous peoples are understood as an integral part of their cultures, spirituality, and economies and also contribute towards the right to self-determination" [107], which is now incontestably recognised by international human rights law [171]. The central role of the land in biocultural construction was confirmed by the Colombian constitutional court in the Terra Digna (or Atrato River) case [106,172,173], the first to protect IPLCs on the basis of biocultural rights. The court indicated that biocultural rights are "rights that ethnic communities have to administer and exercise autonomous guardianship over their territories—according to their own laws and customs—and the natural resources that make up their habitat, where their culture, their traditions and their way of life are developed based on the special relationship they have with the environment and biodiversity" (see Tierra Digna case [174] para. 5.11).

This decision was remarkable in that it underlined, as Posey had already done so well, the importance that control over the land represents for maintaining culture, worldviews, and ways of life, which is then reflected in a certain number of customary laws that govern relationships between humans and the nonhuman [106]. In Madagascar, this can be clearly seen through the concept of *tanindrazana*, which literally refers to the land of the

ancestors, in which Malagasy identity takes the form of a sacred conception of the land and territory [175]. However, none of this appears in the protocols.

In Analavory, the group analysis certainly revealed that, during negotiations around the content of the BCP, discussions had taken place around the relationship to the land and territory. This was not, however, reflected in the protocol. The communities thus mentioned several times their concerns with regard to mining resources in what they considered to be their territory, and which were accessed using explosives, which is counter to the local *fady* in the summer and contravenes local farming systems (the *fady* is based on the belief that explosions attract lightning).

Meanwhile, in Mariarano and Betsako, the PAGE/GIZ technical assistant clearly stated to us on the final official day of the PAGE programme in 2020, that the central issue of the relationship between local communities, their land, and their territories remained to be addressed.

The gap between this and the biocultural approach can be seen in the lack of attention paid to the question of customary rights and local institutions. The Mariarano BCP that, as we have seen, was based on the transfer of management rights, clearly makes references to the *dina*, in particular the "*dina Boeny Miray*" (Community charter, approved by the *Tribunal de Première Instance* of Majunga in 2016, which content focuses on three key points: maintaining social security, preserving the environment, and health and social relations) and the "*Gelose dina*" for the resolution of internal conflicts. However, nothing at all is said about the "*endogenous dina*". This is, nevertheless, a central part of the organisation of community life and social reproduction. As Bérard recalls, it is the "local convention used by the populations with a view to social cohesion, mutual assistance and security, falling within traditional law as it is transmitted orally, pre-dates European colonisation and does not depend on the State for its formation, operation or legitimation" [63]. In contrast, the "Gelose *dina*" are based on local law [176], i.e., a law that is strongly influenced by the state but which leaves scope for local authorities to play a role in the application of norms from a point of view of decentralisation [63]. The Gelose *dina* are not the "sacred agreement" which Weber (1994) [177] saw in the endogenous *dina*, because from the start this involved only keeping the "ritual" part to increase the effectiveness of resource management by regulating access to the extraction of resources [63]. The endogenous *dina* is therefore an instrument led by institutions, and it is not surprising that the Mariarano and Betsako BCP was closely moulded around this logic, far from the traditionalist inspiration so characteristic of the endogenous *dina*.

## 4. Discussion

The first part of the discussion looks at the process of developing the BCPs by trying to understand what hindered its smooth operation, particularly when it came to the issue of taking into account bioculturality. The second part takes a critical look at the issues to be addressed for a better consideration of the biocultural dimension in future Malagasy BCPs.

### 4.1. Factors That Hindered the Development of the BCPs

In Mariarano and Betsako and Analavory, one common observation was the shared basis for development through a so-called "bottom-up" process, which in reality was almost always initiated by external actors [178]. It is certainly true that in the context of the development of BCPs, Natural Justice in particular strove to establish something that looked like "boundary-work", i.e., mechanisms and arrangements that would ensure the "creation and transformation of boundaries between different social worlds that are inhabited by specific communities of actors" [179], in such a way as to overcome the epistemological problems of communication between knowledge systems, but above all to welcome and reconcile any ontological differences that may underline them [180]. Precise examples are the role-playing and sketches developed by Natural Justice through which local communities in the two areas put themselves in the shoes of the negotiators being confronted by the bioprospectors. A significant number of workshops involving small

group work were also organised by Natural Justice. In the case of Mariarano and Betsako particularly, multi-actor workshops were also held, bringing together representatives of the local communities, representatives of Tafo Mihaavo in the Boeny Region, private operators, and various different technical representatives from the regional environmental, agricultural, industry, and trade departments [70].

However, despite all the efforts that were made, our analysis showed that the initiative and influence of external actors, through the very effect of power relations inscribed in the development processes and the way in which the "aid chain" [181] operated, ended up acting like an "'intimate government' which invaded the political imagination and made local 'subjects' pursue goals that they imagined as their own" [182,183].

This is clear in Mariarano and Betsako where the BCP ended up fully espousing the institutional logic of the Gelose, to the point that it was reduced to an economic and institutional incentive resolutely focussed on the sustainable exploitation of local resources (in particular the *Motrobe* value chain), while keeping rural development in its sights.

The Analavory case reveals a similar dynamic, with the project holder (Bioversity International) immediately setting the BCP within the context of the "Grand Bargain" [184] at the heart of ABS and bioprospection. The connection of the Analavory BCP to the ITPGRFA multilateral system, justified by the essential agricultural dimension of the site, nevertheless created an initial blockage effect, because the Natural Justice facilitator was inclined, due to their training and culture, to develop the BCP within the context of the Nagoya Protocol; and all their efforts therefore intended to prepare communities to play a role of supplying resources to the global markets of genetic resources. This situation gives rise to two comments: the first is that, regardless of the aim of actors outside the BCP, in both these cases it involved connecting communities with networks of global trade in resources and endowing them with the role of "environmental entrepreneurs" [185] by playing on "institutional incentives". The second reveals a transformation in development brokerage [186]. Brokers certainly had the "capacity to mediate between several worlds connected to [their] two-fold grasp of the local context and the language of the financial institutions" [187] but they are now increasingly internationalised actors (as Bioversity International in this case) and their "local competence is not really connected to any given area" [187]. They therefore have to go through "assistance chains" and particularly complex networks [182]—which make it possible to deploy alternative objectives and representations [90], a phenomenon which is very clear in Analavory, with "blocking" effects which our work revealed.

*4.2. Putting the "B" Back into the Community Protocols: Where Do We Go from Here?*

As we have seen, the biocultural approach to conservation has found a role in Madagascar, whether within the ABS context (Decree 2017-066 of 31 January 2017), which at least indirectly sanctions the BCPs or the Tafo Mihaavo initiatives around ICCAs. In both cases, the integration of "bioculturalism" within global networks is very clear and potentially of great political importance, as the support that transnational alliances can contribute to the field of conservation in the longer term is well known [154]. However, it is no less true that, at the time of writing, the two pilot protocols examined are still far from keeping their promises, whether that be through recognition of the holistic way of life of the local communities or highlighting all the rights (including the right to land and the right to normative authority), recognition of which would be necessary to maintain communities' "stewardship practices" [18]. Formulated in a deliberately provocative manner, the question becomes: where is the inalienable part of TK in the Malagasy BCPs? What should be done to ensure they effectively support the idea that TK is "a holistic concept that reflects the cultural relationship of ILC [Indigenous Local Community] with their land and natural resources and cannot be separated from cultural identity" [188]? How can we go beyond the "Grand Bargain" narrative articulated around the principle that "( … ) TK might ( … ) help to finance biodiversity through its commercialisation"? The strength of this narrative should be recalled, because it is one of the pillars of the CBD.

However, the Nagoya Protocol and recent decisions at the COP CBD also show that this is not the only possible narrative and that it is in conflict with the representation of IPLCs as "stewards of biodiversity" which has been strengthened by the ethic of stewardship over the past decade.

In Madagascar, the way ahead is narrow. Rebuilding the "total mosaic of a community life that is fragmented under different laws and policies [ . . . ]" [18] must first confront the tendency of the Malagasy state (as is the case for many other states), its officials, and also NGOs operating on the ground, to "divide up" into fixed categories the activities which in reality form a continuum of local practices within worldviews which have nothing in common with the western ontological matrix [189,190] upon which international law on biodiversity is based. There is certainly a normal effect of adjusting to the western legal categories, which no legal–political system can avoid. However, this effect is aggravated in Madagascar for reasons relating to:

i.   A fairly limited consideration of cultural heritage;
ii.  A lack of alliances between activists and academics on questions of defending the rights of farmers and local communities, in contrast to what takes place in Asia and Latin America;
iii. The absence of an overarching legal framework on genetic resources and therefore a partitioning of issues related to genetic resources and those related to PGRFA. This partitioning is clear through the distribution of the various ministries involved in this category of resources, such that the PGRFA are managed by the Ministry of Agriculture and genetic resources other than PGRFA are managed by the Ministry for the Environment, Ecology and Sustainable Development with very little interaction between the two ministries; and
iv.  A lack of state continuity. This is principally due to the recurring political crises that Madagascar has known. Thus, with each change of regime, changeovers in staff in the public administration led to changes in points of contact for local communities and partner NGOs (for more details, see [191]).

The lack of understanding of local practices and vernacular categories became very clear during our ethnographic study in Analavory around the separation of genetic resources and PGRFAs. For the project holder and the facilitator, these categories are clear, because they stem from the international regime on biodiversity. They are reproduced as such in the BCP, which aims, on the one hand at PGRFA and on the other at "genetic resources **other than** Phytogenetic Resources for Food and Agriculture (PGRFA)" (emphasis added). For farmers, however, these categories are foreign to their worldviews and to their praxis, as shown by a long discussion (which cannot be reproduced here) around the word "*fototarazo*", translated into English as "genetic resources". Even when translated into Malagasy, it becomes clear when talking to farmers that the term retains its "extraneity", its radical "otherness": "When you talk to me about *fototarazo*, it's as if you are speaking Greek [French]. I call it traditional culture [ . . . ]" said one farmer (in Malagasy, the expression *fototarazo famboly sy fanao sakafo* incorporates an understanding of both flora and fauna. Another common expression, *harena voajanahary*, also refers to "natural riches" but those that are created by the gods/God. This is the expression used in the Malagasy title of the Protocol).

Obviously, the facilitator did not spare any effort to try to make it understood that traditional seeds were only the first of two facets of the *fototarazo* (or genetic resources *lato sensu*). The (traditional) seeds—i.e., the plants that they cultivate—are effectively only the *fototarazo famboly sy fanao sakafo* (PGRFA). There are also "genetic resources other than" the PGRFA, in other words, everything to which the Nagoya Protocol relates. However, the lack of communication between the worlds can be clearly seen: the farmers are well capable of "filling" this other category, but in a way that cannot satisfy the classifications of international law; the "resources" that the farmers identify are, indeed, nearly systematically referred to as "traditional knowledge". Also, during the group analysis, the "genetic resources other than" the PGRFA become "medicines", "traditional medicines",

especially as one farmer said, "this concerns people who have healing powers", in other words the *mpitaiza olona* (healers) or *mpanao tambavy* (traditional phytotherapists) [192]. The consequence of this is that the vernacular definition of "traditional knowledge" cannot be superimposed upon legally accepted categories (because "plants" are "knowledge").

It is absolutely essential that greater attention is paid by developers and Malagasy officials to the issue of vernacular categories, being particularly sensitive to the often indissociable links between "resources" and "knowledge" and also the way in which each type of resource takes shape in daily practice and the way in which customary rules and local institutions distribute roles around resources depending on status. The example of "*sakafo masaka, tsy mba manan-tompo*"—"prepared food has no owner" recalls with particular acuity, that some institutions, such as a written contract or the "disembedded" market [193], are quite simply incompatible with the "ontological" status of certain "goods" (which goes back to the issue of the "inalienable" part of heritage).

Another significant challenge is around granting greater normative and institutional autonomy to local communities, the condition sine qua non to guaranteeing the maintenance of cultural practices and worldviews linked to sustaining biodiversity. In reality, what we see in the Malagasy case is the state's strong will to control, if not the resources themselves, then at least the local communities which manage them—illustrating what has been described in environmental conservation as a shift from control over nature towards the control of communities [160]. This obviously reveals the fear of losing, for a second time, control over "nature" (the first being with the decentralisation following the Washington Consensus); at the same time as the states' tendency—which is the mark of high-modernism according to James Scott—to roll out schemes that aim to "make legible", i.e., "make uniform, standardise and simplify" in order to better control [194]. "Making legible" appears to have been a motivating factor for the drafters of the Gelose Act, following a spectacular U-turn which continues, 30 years later, to produce effects on policies to manage natural resources. While the initial discussions around Gelose focussed on the need to "be based on the *Fokonolona* to transfer the management of natural resources, the supreme bodies of traditional local power, which showed that they were essential and were the most obvious and least conflictual point of reference in terms of the natural management of local resources ( . . . )"; a last-minute change gave rise to a "basic local community" translated into Malagasy as *Vondron'Olona Ifotony* (VOI) [63]. The desire to control and the correlated search for "legibility" are all the more doubtful as this U-turn was justified by the "fear of officials from the technical departments of losing part of their power to the *Fokonolona*, as well as by the very essence of the *Fokonolona* themselves—entities which were fluid and difficult to grasp by their very nature [63]."

The current BCPs underwent the same "formatting" dynamic, because in both Mariarano and Betsako and Analavory, the BCPs fell within local structures which were in the state's favour and, above all, with legal status: the VOI for the first and the Seed Production Cooperative for the second. These approaches certainly facilitated the process of negotiating the BCPs insofar as, firstly, their promoters were able to build upon what already existed, without dealing with the delicate question of the outline of the community or communities; and then the "legally constituted entities" enabled the traceability of funds or benefits that accompanied the bioprospection contracts.

The opposite is clear: what does the imposed entity represent with regard to the complexity of the very concept of "community", the complex and moving interplay of affiliations which marks life and death, its boundaries which open and close according to the goods to be exchanged and the questions to be settled [98,195,196]? These games of boundaries, outlines, and scales are visible in Analavory, the BCP of which is aimed at the rural commune of Analavory, while only three *fokontany* were involved. Upon even closer inspection, the FaMA cooperative appears to have kept control over the process of developing the instrument. See also the hesitations of the ministries when it came to the appropriate scale (VOI, *fokontany*, or commune?) upon which to negotiate the BCPs—and which says a lot about the state's unease towards the community issue. As Cori

Hayden recalls, bioprospection calls for a "taking back", a "return" which "cannot proceed, unleashed and unchaperoned, directly to market". This "return" requires a destination that cannot be the individual. As she states, " [ . . . ] the individual is a nervous-making entity: a conduit to the specters of property rights, commodity exchange, and 'indue inducement'" [197]. Benefit sharing necessarily requires, "something like 'community'", a "collective"—a collective that the state, as well as bioprospectors, continue to try to define in simple and homogenous terms, or at least according to criteria that do not risk bringing out the (biocultural) complexity of what has been taken (knowledge and resources) on the ground. The market also calls for "legibility". From the point of view of the communities, this political–ontological reduction is obviously problematic because, in addition to generating conflicts, it brings back into question the local rules on matching and affiliation and, with them, the entire network of social and cultural norms that govern the management of resources (particularly rules around seed exchanges) [81].

## 5. Conclusions

What this paper shows is that BCPs can be powerful tools for protecting and revital-ising TK, but that this is only the case if the holistic nature of TK is taken seriously, i.e., TK as a biocultural whole, bound together with the territory, local customs, and biological resources [1,2]. In this light, a BCP cannot simply be a tool for institutional steering of re-source management, a "grammar of institutions" [198] intended to strengthen the collective action of communities and facilitate their insertion in global exchange markets. Decision makers and experts should pay as much attention as possible to vernacular categories and norms (i.e., "vernacular law" such as local taboos (*fady*) as well as traditional *dina*), by promoting the normative and institutional autonomy of communities—which is supposed to be one of the pillars of the BCP.

The importance of really placing TK within its complex biocultural nexus and not skewing the meaning of "biocultural" should not be overstimated; particularly at a time when "biocultural approaches" to conservation [199,200] and "biocultural-based" policy or ethical tools [201] proliferate, including in the "Global North" as the 2019 Atateken North American Regional Declaration on Biocultural Diversity and Recommended Actions [202] or recent initiatives around the Hawaiian "ahupua'a system" [203] or ancestral agricultural practices known as "āaina malo'o", still in Hawaii [204], testify. Otherwise, biocultural approaches—of which BCPs are part—risk being no more than a tool aiming to replace real political negotiation through managerial interventions [183].

**Author Contributions:** Conceptualization, M.R. and F.G.; methodology, M.R. and F.G.; validation, M.R. and F.G.; formal analysis, M.R. and F.G.; investigation, M.R. and F.G.; data curation, M.R.; writing—original draft preparation, M.R.; writing—review and editing, M.R. and F.G.; visualization, M.R.; supervision, F.G.; project administration, F.G.; funding acquisition, F.G. All authors have read and agreed to the published version of the manuscript.

**Funding:** This research was carried out as part of the "BioCulturalis" project funded by the ANR (No. ANR-18-CE03-0003-01) and managed by F. Girard.

**Institutional Review Board Statement:** Ethical review and approval were waived for this study, since no questions of a personal or sensitive nature were asked.

**Informed Consent Statement:** Free, prior and informed consent was obtained from all subjects involved in the study for semi-directed interviews, group analysis (Analavory), ethnographic work-shops (Analavory), community workshops (Mariarano), multi-actor workshops (Mariarano), as well as for audio recording of verbal exchanges and interviews. In transcripts of recorded interviews or workshop subjects are given aliases and code lists are stored separately from transcripts and are only accessible by the authors.

**Data Availability Statement:** Data is not publicly available. Audio recording, digitalised fieldnotes and interview transcripts can only be accessed by the authors.

**Acknowledgments:** We would like to kindly thank the following people for their valuable collaboration in our investigations (acknowledgement is in alphabetical order): M. Andriamahazo, S. Rambinintsaotra, L. Ramamonjisoa, H.F. Ranaivoson, J. Rasoloaraijaona, and M. Razafindralambo. All members of the community of Analavory are also warmly thanked for their hospitality, warmth, and willingness to allow us to join in on their daily lives. We would also like to express our sincere thanks to all the PAGE/GIZ teams for their support and sharing of experiences on the development process of the Mariarano and Betsako BCP.

**Conflicts of Interest:** The authors declare no conflict of interest.

## Appendix A

**Table A1.** Summary of interviewees and information collected.

| Categories of Stakeholders | Stakeholders | Number of Respondents | Main Themes Discussed |
| --- | --- | --- | --- |
| Project holders | Bioversity International | 1 | Origin of the BCP |
| | GIZ | 1 | Negotiation and development process |
| Facilitator | Natural Justice | 1 | Origin of the BCP Negotiation and development process Impact of the BCP on the communities in Analavory and Mariarano/Betsako |
| Ministry representatives | MAEP (Ministry of Agriculture)—ITPGRFA National Focal Point | 1 | Development process of the BCP of Analavory Ministry's view on farmers' rights under the ITPGRFA |
| | MEDD (Ministry of the Environment)—ABS National Focal Point (the former focal point and the current one) | 2 | Development process of the two BCP Regulatory framework for BCPs in Madagascar Ministry's view on the rights of local communities and the ABS framework |
| | FOFIFA—National Centre for Applied Research in Rural Development | 1 | Farmer education and outreach (seed, marketing, and distribution channels) |
| | Official Seed and Plant Material Control Service (SOC) | 1 | Seed marketing and seed certification regulations in Madagascar |
| | Regional technical staff (Boeny and Itasy) | 3 | Development process of the two BCPs |
| ILC representatives | Tafo Mihaavo | 3 | ICCAs-territories of life in Madagascar Community-based natural resource management (issues related to the upkeep of customary governance systems) |
| | Members of the FaMA cooperative (Analavory) | 25 | Views on the development process of the BCP of Analavory and its impact on the on the communities and on seed exchanges |
| | Other member of the community (Analavory) | 10 | The Ray aman-dReny (village elders) (questions on values, customs, representations of nature, and ways of life) |
| Private sector | Bionnexx | 1 | Approaches for collecting genetic resources at the local community level (e.g., are there contracts with the central ministry? How are benefits shared with communities?) |
| | Homéopharma | 1 | |
| Other | OMAPI—Malagasy Industrial Property Office | 1 | Legal framework on intellectual property rights |

## Appendix B

**Biocultural Community Protocol of local communities custodians of biodiversity and holders of traditional knowledge of communities**
*Tari-dalana sy Vina Iombonana (TVI) Fokonolona mpiahy ny harena voajanahary sy mpanana ny fahalalana nentim-paharazana ao Mariarano- Antanandava – Komamy – Ankilahila – Marosakoa – Tsakoambezo - Tsianinkira*

**Preface**
- Presentation of the BCP as a common vision on benefit sharing from the perspective of the conservation and sustainable use of biodiversity

**Introduction**
- A single BCP for seven sites
- Texts governing the development of BCP at national and international level

**I. Organisation of decision-making**
- Importance of community meetings for decision making
- Main issues for community decision making
- Form of community meetings

**II. Method for granting prior consent for the issuance of a permit**
- Steps to be taken to grant prior consent if it is for a major piece of research or for the use of research
- Steps to follow if it is a piece of student research

**III. Conditions of access to Traditional Knowledge**
- Access condition if one person holds the TK
- Access condition if a group of people holds the TK
- Need for a written contract in the presence of the President of the Fokontany or the sector, the BCP committee and a representative of the Competent National Authority or its regional branch

**IV. Purchase of Biodiversity Products**
- Need to obtain the legal permission of access (ABS permit or collection permit) before purchasing resources
- Need to state the price agreed by all parties and the collection location in the contract.

**V. Local community engagement for sustainability and the use of biodiversity**
- Community commitment for biodiversity conservation: period, quantity and method of harvesting
- Respect for the working and operating area
- Collaboration for biodiversity conservation

**VI. Traditions and values**
- Loss of traditional values nowadays
- Need to approach a village elder to find out more about customs and traditions
- Incentives to respect customs and taboos

**VII. Benefit sharing arrangements**
- Characteristics of shared benefits on the use of biodiversity
- Benefits related to the use of TK
- How benefits are shared

**VIII. Conflict resolution**
- Steps to follow if conflicts arise within the community: amicable settlement → application of the Dina (VOI and Boeny Miray) → in case of failure, resort to "higher bodies".
- If conflicts arise within the local community, with the business operator or researcher: immediate cessation of operations → application of the Dina (VOI and Boeny Miray) → in case of failure, resort to "higher bodies".

**Visual summary of the main principles of the BCP (only appears in the Malagasy version)**
- Summary of I to VIII

**Postface**
- Reminder of the objective of the BCP: to have a common vision, a common working method in the seven sites for the sustainable use of biodiversity and TK
- BCP as a communication tool for external actors: private and public sectors, research institution and organisations, to inform them on the different rules and principles to be respected

**Table of contents**

**Figure A1.** Summary of the content of the BCP of Mariarano and Betsako.

**Biocultural Community Protocol of the peasants of Analavory on Access to and the Sharing of Benefits arising from the Use of Genetic Resources and Associated Traditional Knowledge**
*Arofenitra iombonan'ny tantsahan'Analavory : Tari-dalana amin'ny fangalana sy fifampizarana tombon-tsoa azo amin'ny fampiasana ireo harena voajanahary sy ireo fahalalana nentim-paharazana mifandraika aminy*

**Preface**
- Presentation of the community, the wealth of its natural resources;
- Definition of the Protocol: "guide" to strengthen and inform about the local organisation for the conservation, sustainable use, exchange and sharing of benefits arising from the use of genetic resources and associated traditional knowledge;
- Origin of the Protocol;
- Addressees of the Protocol;
- Aims of the protocol;
- Reminder of the national and international instruments supporting the implementation of the BCP;
- Protocol outline.

**I. Methods genetic resources management by the farmers of Analavory**
- References to the "Community Register of Biodiversity" and the seed bank
- References to the holders of TK. Role of the "*Ray Aman-dReny*" (ancestors) and the "*mpanao tambavy*" (traditional phytotherapists).

**II. Challenges encountered in the management of genetic resources and TK associated in Analavory**
- Disruption of the cropping calendar due to the degradation of the environment in general; disappearance and rarefaction of animal and plant species in Analavory;
- Forgetting and loss of TK held by the *Fokonolona* due to the modernisation of medicine. Loss of TK associated with resources due to the rarefaction of these resources;
- Decreasing interest of the youth in TK;
- Discrepancy between Analavory's local resource potential and the low standard of living of peasants;
- Problems linked to the involvement of farmers in decisions that may affect their lives (practices of exploitation of local resources at odds with local customs)

**III. Modalities of exchanges of Plant Genetic Resources for Food and Agriculture (PGRFA) between farmers in Analavory**
- Peer-to-peer seed swapping as the main modality of seed exchange;
- Evocation of "*firaisan-kina*" (solidarity) as a fundamental basis for exchange and the role of plant genetic resources in everyday social life.

**IV. Modalities of exchange of, and access to, PGRFA from farmers outside the commune or abroad**
- Role of the SML;
- Role of the general assembly of Fa.M.A. members and the rest of the community in decisions on the inclusion of a resource in the Multilateral System;
- Steps to apply for PGRFA from outside (Multilateral System or other research centers): mention of participatory plant breeding.

**V. How to apply for genetic resources located in Analavory**
1. Prior and informed consent of the entire community for access to genetic resources and associated TK;
2. Steps to be taken when using resources for research purposes or when using resources and associated TK for commercial purposes;
3. Description of the 6 steps to be followed for access and benefit sharing.

**VI. Preservation and development of TK**
- Recognition of the link between the preservation of TK and the preservation of genetic resources;
- Reminder of the conditions of access to TK (mutually agreed terms and benefit sharing)

**VII. Benefit sharing**
- Monetary and non-monetary benefits;
- Preference is given to benefits that promote the agricultural sector, strengthen the production capacity of farmers or increase the competitiveness of products on the market.

**VIII. Resolution of possible conflicts and grievances**
- Preference is given to amicable settlement;
- In case of failure, resort to "higher instances".

**IX. Call for the respect and promotion of the rights of local communities and farmers**
Reminder of the international and national legal framework:

1. International Treaty on Plant Genetic Resources for Food and Agriculture (ITPGRFA) and Order No. 11.567/2010 on interim measures related to requests for access to PGRFA and benefit sharing within the framework of the Multilateral System of the ITPGRFA;

2. Nagoya Protocol on Access to Genetic Resources and the Fair and Equitable Sharing of Benefits Arising from their Utilization, and Decree n°2017-066 of 31 January 2017 on the regulation of access and benefit sharing arising out of the use of Genetic Resources;

3. Texts governing TK: Art. 8(j) of the Convention on Biological Diversity, Decree 2017-066 of 31 January 2017 on the regulation of access and benefit sharing arising out of the use of Genetic Resources and Law 2013-017 of 12 December 2013 for the safeguarding of national intangible heritage;

4. United Nations Declaration on the Rights of Peasants and Other People Working in Rural Areas (art. 3.5; 5 and 10)

**Signature page**
1. Top of the page: Logos of the two ministries: Ministry of Agriculture and Environment.
2. Signatures and stamps of the Director of the Regional directorate of Agriculture and Animal Husbandry for the Itasy Region, the Director of the Regional directorate of Environment, Ecology and Forestry for the Itasy Region, and the Deputy Mayor of the Rural Commune of Analavory, dated 27/12/2017;
3. At the bottom of the page: logos of Darwin Initiative, Bioversity International and ABS Initiative.

**Last page**
1. Contact of individuals and their institutions in case of questions related to the use of the PBC.

**Figure A2.** Summary of the content of the BCP of Analavory.

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
