# Peer review of "Protecting Traditional Knowledge through Biocultural Community Protocols in Madagascar: Do Not Forget the “B” in BCP"

_sustainability, doi:10.3390/su131810255_

Round 1

Reviewer 1 Report

I really enjoyed reading this article. A lot of work went into it and it has interesting theoretical aspect while practical. Right now this is way too long 40+ pages. I think much of the information could be summarized. Much of the technical jargon could be removed from this paper. I would think that the Sustainability audience is more of a general audience, not in your specifi field. To make this appeal to more people you also need to make the article relevant to different geographies. For example, as a researcher living in the U.S. I am not sure how the topic affect us here. 

Reviewer 2 Report

Sustainability 1338675

Protecting Traditional Knowledge through Biocultural Community Protocols in Madagascar: Do Not Forget the “B” in BCP

Comments:

The research studies the two indigenous communities in the Madagascar to bring a critical picture of biocultural knowledge conservation through understanding the different worldviews, values and lifestyles of the local and indigenous people. The article is interesting with sufficient details and fit for the journal Sustainability. Especially I liked the stress on biocultural part for sustainable resource management beyond economic incentives and other approaches that can narrow the many and dynamic ways through which people and nature mutually interact through culturally embedded meanings and values.

However, the article still has rooms for improvement. Please see my comments and suggestions below for more details:

Style:

The authors have used a lot of footnotes, which makes the work difficult to follow. My suggestion is that the authors need to find a way to include these footnotes in the main text more, while reducing the words used in the footnotes.

There should be a description of the study area. The study area should especially introduce briefly about the biocultural characteristics (biodiversity including endemic wildlife, traditional people and their relation with the biodiversity through culture and knowledge).

Please provide a short description of the customary laws that are key to conserve biodiversity in the case study areas. If it is not possible to include this in the main text, the authors can provide this in the appendix section, while cross-referring this in the main text when needed.

Methods:

The authors need to structure the methods more. The steps taken to gather and analyze data should be clearly written. Mention the number of interviews conducted (provide a table that shows the background and expertise of the interviewees). I think it was more unstructured and the authors followed a discussion style in the interviews, but please mention the main threads of question asked (if possible provide a sample of interview questions in the Appendix). 

Similarly, please explain in detail about the workshops conducted. If possible provide a socioeconomic profile of the workshop participants in a table. Explain how you gathered the workshop participants where and how the workshops were conducted (i.e., the deliberation process),  and were there repeat in the participants who attended the workshops. More details should be provided about the deliberation process especially to show how it enabled the authors to understand the values, worldviews and lifestyles that are backed by TK. This is not easy to do and thus these information will be of good importance for both the internal validity of the research as well as possible replication by other scholars in other similar studies. 

Line 994-995>> "A significant number of workshops took place involving small group work." Please explain these group works and why these were not carried out in all the workshops? How did this may have or may have not affected the overall results?

I will appreciate if the authors can provide some pictures of their field study (the manuscript has very few figures at present). Also it will be very good if the authors can produce some interesting visualizations through table/s and figure/s (For example, the worldviews, values and lifestyle of the TK holders that bind the socioecological system together, and the main components of the protocols that can erode these TK based understanding and use.)

The article is well written but is very long. I think the authors need a way to reduce the words to some extent for better readability especially in the results and discussion part.

Conclusion:

The article does not provide a conclusion section. Please provide a conclusion section summarizing the study while adding the novetly of the research.  

Reviewer 3 Report

The present article is approaches topics like traditional knowledge and biocultural community protocols aiming to identify how the traditional knowledge can be revitalize in countries like Madagascar.

The abstract is well developed, provides significant information on the article. However, it is not cleat from this part which is the main conclusion of this work. This aspect should be corrected.

The introduction approaches a wide range of concepts, elements etc. and should be better structured in order to focused only on the relevant topics. Please also emphasize which are the main questions / research gaps that will be approached in this work.

It is not clear which are the main resources that will be used in this article (materials/ methods). How did you select the case studies and why? Which are their main and relevant characteristics?

Please synthesize and emphasize the main results of your work. 

Which are the main conclusions of your work? Please emphasize them in a separate section.

Overall, the article requires a major revision.

Round 2

Reviewer 1 Report

Even when the article is still very long (8,000 might be the norm), I am satisfied with the changes, especially the introduction, which gives good background.  

Author Response

Thank you for proofreading the article.

Reviewer 3 Report

No other comments to make.

The article can be accepted as it is.

Author Response

Thank you for your proofreading.